# Robust global detection of forced changes in mean and extreme precipitation despite observational disagreement on the magnitude of change

Iris Elisabeth de Vries[1], Sebastian Sippel[1], Angeline Greene Pendergrass[2,3], and Reto Knutti[1]

[1]Institute for Atmospheric and Climate Science, ETH Zurich, Zurich, Zurich, Switzerland
[2]Earth and Atmospheric Sciences, Cornell University, Ithaca, NY, USA
[3]National Center for Atmospheric Research, Boulder, CO, USA

**Correspondence:** Iris de Vries (iris.devries@env.ethz.ch)

**Abstract.** Detection and attribution (D&A) of forced precipitation change is challenging due to internal variability, limited spatial and temporal coverage of observational records, and model uncertainty. These factors result in a low signal-to-noise ratio of potential regional and even global trends. Here, we use a statistical method — ridge regression — to create physically interpretable fingerprints for detection of forced changes in mean and extreme precipitation with a high signal-to-noise ratio. The fingerprints are constructed using CMIP6 multi-model output masked to match coverage of three gridded precipitation observational datasets – GHCNDEX, HadEX3, and GPCC –, and are then applied to these observational datasets to assess the degree of forced change detectable in the real-world climate in the period 1951-2020.

We show that the signature of forced change is detected in all three observational datasets for global metrics of mean and extreme precipitation. Forced changes are still detectable from changes in the spatial patterns of precipitation even if the global mean trend is removed from the data. This shows detection of forced change in mean and extreme precipitation beyond a global mean trend is robust, and increases confidence in the detection method's power, as well as in climate models' ability to capture the relevant processes that contribute to large-scale patterns of change.

We also find, however, that detectability depends on the observational dataset used. Not only coverage differences but also observational uncertainty contribute to dataset disagreement, exemplified by times of emergence of forced change from internal variability ranging from 1998 to 2004 among datasets. Furthermore, different choices for the period over which the forced trend is computed result in different levels of agreement between observations and model projections. These sensitivities may explain apparent contradictions in recent studies on whether models under- or overestimate the observed forced increase in mean and extreme precipitation. Lastly, the detection fingerprints are found to rely primarily on the signal in the extratropical Northern Hemisphere, which is at least partly due to observational coverage, but potentially also due to the presence of a more robust signal in the Northern Hemisphere in general.

# 1 Introduction

Precipitation changes may be among the most important consequences of anthropogenic climate change. Yet, robust detection and attribution (D&A) of forced change in the water cycle is impaired by low signal-to-noise ratios. The concept of detection and attribution is to use climate model simulations in which the applied forcings are known and internal variability can be reduced by averaging multiple realisations, to estimate a so-called fingerprint that represents the effect of the applied forcings on climate variables of interest. Subsequently, the degree to which this fingerprint can be detected in observations is assessed; if the fingerprint signal is significant and in agreement with the models, the forcing signal is said to be detected and attributed to the applied forcings. The low signal-to-noise ratios of precipitation D&A result from many factors (Balan Sarojini et al., 2016). First, internal variability of precipitation and related processes is high (Deser et al., 2012; Hoerling et al., 2010; Balan Sarojini et al., 2012). Second, models show relatively large disagreement in water cycle simulations due to, for example, structural uncertainties such as parametrised convection, and differing climate and hydrological sensitivities (Pendergrass, 2020). There can also be discrepancies between model representations of the water cycle and observations (Mehran et al., 2014; Wehner et al., 2020). Lastly, signal robustness suffers from limited spatial and temporal coverage of observations, and biases can be introduced by changing coverage and station density over time, as well as gridding procedures (Balan Sarojini et al., 2012; Dunn et al., 2020). Here we present a detection method based on regularised linear regression – ridge regression – that is suitable to detect forced changes in global metrics of mean and extreme precipitation with high signal-to-noise ratio, despite the listed challenges.

Models and observations roughly agree on a rate of specific humidity increase with global mean temperature of $\approx 7\%K^{-1}$, following theoretical relationships (Held and Soden, 2006; Dai, 2006). Extreme precipitation scales approximately with this rate of increased precipitable water and increases over most of the global land, albeit atmospheric dynamics modulate the increase in some regions (O'Gorman and Schneider, 2009; Fischer and Knutti, 2016; Pfahl et al., 2017). Changes in global mean precipitation are associated with the atmospheric energy balance, resulting in a smaller increase of $\approx 1\text{-}3\%K^{-1}$, with an underlying spatial pattern of hydrological cycle intensification (Allen and Ingram, 2002; Allan et al., 2014; Pendergrass and Hartmann, 2014; Douville et al., 2021). Changes in mean precipitation over land are not well described by this pattern intensification, though, and are expected to be lower and more complex due to effects of water availability and relatively higher warming rates over land compared to oceans (Douville et al., 2021; Byrne and O'Gorman, 2015; Roderick et al., 2014). Besides local climatology, changes in factors such as large-scale atmospheric circulation, water availability and the vertical structure of the atmosphere play a role in the local precipitation response to forcing (Byrne and O'Gorman, 2015; O'Gorman and Schneider, 2009; Pfahl et al., 2017).

For mean precipitation, anthropogenically forced changes have been detected and attributed on a global land level and for regions defined by latitude bands (Fischer and Knutti, 2014; Knutson and Zeng, 2018; Noake et al., 2012; Polson et al., 2013; Marvel and Bonfils, 2013). Anthropogenic aerosols and GHGs have opposing influences on the hydrological cycle (Wu et al., 2013; Bonfils et al., 2020; Salzmann, 2016), implying that continued increase of GHGs and decrease of aerosol emissions will lead to stronger GHG signatures in mean precipitation. Although studies agree on the presence of a signal in observations,

they disagree on the strength. Models have been suggested to overestimate (Fischer and Knutti, 2014) as well as underestimate (Noake et al., 2012; Wu et al., 2013; Polson et al., 2013; Knutson and Zeng, 2018) observed trends.

For extreme precipitation, optimal fingerprinting methods and spatial aggregation approaches have led to detection and attribution of anthropogenically forced changes over global land and for distinct Northern Hemispheric regions (e.g. Min et al., 2011; Zhang et al., 2013; Fischer and Knutti, 2014; Paik et al., 2020; Kirchmeier-Young and Zhang, 2020; Sun et al., 2022; Fischer and Knutti, 2016). However, for extreme precipitation there is also disagreement regarding the strength of the forced signal in observations. A subset of studies finds CMIP multi-model ensembles generally underestimate changes compared to observations (Min et al., 2011; Fischer and Knutti, 2014, 2016; Borodina et al., 2017), whereas others find the opposite (Zhang et al., 2013; Paik et al., 2020; Sun et al., 2022).

Hence, the degree to which model simulations accurately represent the responses of precipitation relevant processes to forcing, and thus accurately simulate past and future changes in precipitation remains up for debate. Knowledge of the severity of current climate change effects on the water cycle, as well as the congruence of modelled and observed historical forced changes in the water cycle is important for adaptation policies and improvement of future projections.

Recent studies using data-science methods of varying complexity for the purpose of reducing the signal-obscuring effects of uncertainties and internal variability have detected forced signals in temperature as well as mean and extreme precipitation (Sippel et al., 2020; Barnes et al., 2019, 2020; Madakumbura et al., 2021). Here, we show that regularised linear regression can alleviate some of the difficulties in precipitation D&A by reducing the influence of internal variability and structural model error on detection results. We use regularised linear regression to construct high signal-to-noise ratio fingerprints for detection of the forced response in mean and extreme precipitation based on observational coverage, and apply these to several station-based observational datasets to assess whether significant forced changes are detected and have emerged from internal variability. The simultaneous assessment of multiple observational datasets provides an overview of how modelled and observed forced changes compare, and sheds light on the aforementioned contradictory findings. We analyse forced signals in annual precipitation anomalies, and also in anomalies from which the global mean trend is removed, relying on spatial pattern information alone. The latter approach highlights relative regional responses to forcing, and tests whether spatial pattern changes in models are in accordance with observations.

## 2 Methodology

In our detection procedure ridge regression (RR) models are trained on simulated spatial patterns of precipitation with known forcings to determine fingerprints of the modelled forced response of annual mean total precipitation (PRCPTOT) and annual extreme precipitation (Rx1d: precipitation accumulation on the day with most precipitation each year). The fingerprints are such that they predict the global forced response from the spatial locations where observational data are available. The RR-fingerprints are applied to observations to isolate an estimate of the real-world forced response from internal variability. Several data processing and regression steps are needed to achieve this. We describe the general procedures here; supplementary Sect. S1 contains additional details, and Sippel et al. (2020) describes the method used here in detail.

Our method bears similarity to (non-)optimal fingerprinting methods for D&A which have been developed over the past decades. From Klaus Hasselmann's seminal paper on signal-to-noise in detecting forced climate responses (Hasselmann, 1979), optimal fingerprinting (e.g. Hegerl et al., 1996; Allen and Stott, 2003; Ribes et al., 2013; Ribes and Terray, 2013) and detection methods based on pattern similarity (e.g. Santer et al., 1995, 2013; Marvel and Bonfils, 2013) have evolved. In optimal fingerprinting, observations are regressed on a "guess pattern" of the forced response derived from models, using an estimate of internal variability, resulting in scalars ("scaling factors") representing the strength of the guess patterns in observations. Our ridge regression based detection method differs from this approach in that we do not regress observational data on simulated estimates of the forced response, but determine a detection model based on model data only. It is more closely related to pattern similarity methods, where an EOF-based signal pattern is referred to as the fingerprint, and the projection of spatiotemporal observations onto this pattern yields a one-dimensional (temporal) estimate of the forced response in observations. Our method builds on this in a straightforward way by adding a step to optimise signal-to-noise ratio. In our method, we project observations not onto the signal pattern directly, but onto a regression coefficient pattern that "optimally" (linearly, optimised by regularisation) projects simulated spatiotemporal Rx1d or PRCPTOT patterns onto a one-dimensional detection space based on the signal pattern (see Sect. 2.2 and 2.3 for details). Regularisation optimises the regression coefficient pattern for high signal-to-noise ratio across models and thus improves generalisability. The detection metric is then applied to map spatiotemporal observations onto the one-dimensional detection space, thus extracting the forced response signature in the real-world climate.

We believe the advantages of our method lie in (1) its relative simplicity and close links to pattern similarity based D&A methods, while going beyond comparisons to the signal pattern (e.g. Santer et al., 2013; Marvel and Bonfils, 2013; Bonfils et al., 2020) or spatial aggregation techniques (e.g. Fischer and Knutti, 2014; Borodina et al., 2017), (2) the interpretable and intuitive fingerprint (spatial coefficient map) that reflects regions exhibiting high signal-to-noise ratio climate change signals, (3) the fact that the estimate of the observed forced response is a time series, allowing for analysis of trends and (4) the possibility to straightforwardly introduce additional constraints to, for instance, increase robustness of detection with respect to specific climate uncertainties, such as decadal-scale internal variability (Sippel et al., 2021). This method fits in recent developments in D&A that move towards mapping multidimensional data onto a one-dimensional detection space. Studies based on neural networks and deep learning for D&A (e.g. Barnes et al., 2019, 2020; Labe and Barnes, 2021; Madakumbura et al., 2021), employ non-linear methods but use a very similar framework with similar goals. We do not argue that ridge regression is fundamentally better than any of the mentioned methods, but we are convinced that the intuitive, physical outputs combined with high signal-to-noise ratio can be valuable for trend detection and attribution.

## 2.1 Model simulations and observational data

We use model simulations from thirteen CMIP6 models with at least three members with historical and SSP245 data (Eyring et al., 2016). For models that have more than three of such members, the first three are selected. We include a large number of models and multiple members per model to cover as large a range of the physically plausible climate responses as possible. Including multiple members per model accounts for within-model internal variability, and including multiple models accounts

additionally for across-model variability and uncertainty. Furthermore, the high number of models used ensures that one single model with, for example, a deviating high or low climate sensitivity, does not have a strong effect on the combined result. As unforced control data, 450-year piControl simulations for ten out of the thirteen models are available and used, i.e. 4500 years
of unforced data. Long piControl simulations are used to sample a representative distribution of unforced trends, increasing confidence in the assessment of whether forced trends lie outside the likely range of unforced trends. See table S1 in the supplement for an overview of CMIP6 data used.

    In an effort to address observational uncertainty, three observational datasets are used: for Rx1d HadEX3 (Dunn et al., 2020) (1.875°×1.25°, 1951-2018), and GHCNDEX (Donat et al., 2013) (2.5°×2.5°, 1951-2020) are used. For PRCPTOT, HadEX3,
GHCNDEX and GPCC (Schneider et al., 2017) (2.5°×2.5°, 1951-2019) are used. GPCC does not provide extreme indices and is therefore only used for PRCPTOT. All three observational datasets are gridded data derived from station observations (Dunn et al., 2020; Donat et al., 2013; Schneider et al., 2017). HadEX3 and GHCNDEX only provide values for gridcells where three stations are available within the decorrelation length scale of the gridding procedure, leading to spatially incomplete maps. GPCC, on the other hand, interpolates to all land gridcells. In order to create a reliable GPCC record comparable to the other
datasets, we mask this data to include only gridcells in which data from three stations were available, as per the station density data provided in GPCC as well.

    Coverage differs for each timestep within each dataset. In order to generate time-independent fingerprints, we create one single coverage mask for each dataset representative of 1951-present. Gridcells for which a maximum of 3 time steps (years) is missing are included, the missing time steps are set to the time mean of the gridcell in question. This implies that the total
fraction of filled-in datapoints (gridcells×timesteps) ranges from 0.4-0.9 %.

    We note that the nature of model data and observational gridded data differs, as model precipitation values are spatial gridcell averages, whereas methods to grid station-based point observations onto a regular grid result in values representative for gridcell centres (Dunn et al., 2020). This is mainly a result of station observations being too sparse to allow for reliable estimation of gridded area mean values. In addition, gridded observational data for Rx1d is constructed by first extracting station maxima
and then gridding these, creating an aggregated area maximum value (Dunn et al., 2020; Donat et al., 2013). In contrast, Rx1d determined from model data reflects the maximum of the gridcell average precipitation. These structural differences affect precipitation indices, extreme indices in particular, and reduce the direct comparability of model and observational absolute precipitation values. This observation-model discrepancy can be overcome by using daily, gridded observational data such as REGEN (Contractor et al., 2020), but reliable datasets with long records of this kind are not numerous. Trend biases due to these
structural differences have been shown to be negligible, however, justifying the comparison between models and observations made in this study (Dunn et al., 2020; Avila et al., 2015).

## 2.2   Data processing

Rx1d is determined as the maximum daily amount of precipitation per year for each location on the original grid. We regrid modelled PRCPTOT and Rx1d fields onto the grids of HadEX3 and GHCNDEX. This order of operations – first extract
maxima, then regrid – is used to avoid that Rx1d values acquire additional bias due to the spatial averaging involved in

regridding; the used approach is closest to how observational Rx1d indices are computed, as described above. A disadvantage of this order of operations is that annual Rx1d values may occur on different days in neighbouring gridcells, meaning that the regridded values strictly no longer represent the most extreme day per gridcell, but rather a representative local extreme level. This is however also the case for observations.

As the GPCC grid is nearly identical to the GHCNDEX grid, we regrid GPCC to the GHCNDEX grid so that the GHCNDEX-regridded CMIP6 simulations can be used. PRCPTOT and Rx1d annual anomalies with respect to the 1951-2014 reference period are determined per gridcell on the observational grids. For CMIP6 data, anomalies of individual members are computed with respect to the annual mean of the ensemble mean of the model in question. For these ensemble means, all available model members are used to reduce noise where possible, even though only three members are used in the RR model training. This re-
moves potential systematic model biases in absolute precipitation levels, which is required for meaningful prediction of forced trends.

    These fields of single-model ensemble member anomalies, masked to observational coverage, serve as predictors to train the RR model, with the goal of predicting the forced response, and are used as RR inputs to obtain model forced response estimates. The observational anomalies serve as input to the trained RR model to determine the observed forced response estimate. In a
second application of the method, we subtract the masked, area-weighted spatial mean from the predictors and observational data for each time step. These detrended predictors thus only contain the *relative* pattern changes in precipitation.

    The RR model's purpose is to predict the forced response from the predictors (anomaly maps), hence the RR model is trained with a forced response proxy as target variable (predictand). Different RR models are trained for PRCPTOT and Rx1d – each has their own forced response proxy as target variable. In order to include the pattern of change, the main forced
response metric used in this study is based on empirical orthogonal function (EOF) analysis of the unmasked multi-model mean anomaly maps, conceptually similar to a traditional way of extracting forced responses (e.g. Santer et al., 1995; Hegerl et al., 1996). Multi-model mean anomalies are determined by taking the mean of the single-model ensemble means and centring this on the 1951-2014 reference period. By first computing single-model ensemble means, all models contribute equally to the multi-model mean regardless of ensemble size. The first EOF of the data represents the spatial pattern that explains most of
the variance in the data, and its corresponding first principal component is a timeseries reflecting the strength of that pattern in the data. We perform the EOF analysis over the entire length (1850-2100) of the multi-model mean record. The first principal component correlates highly with the area-weighted global mean change; the average Pearson correlation coefficient is 0.9 for PRCPTOT, and 0.99 for Rx1d (see supplementary Fig. S3), and the first EOF pattern is very similar to the linear multi-model mean trend pattern (see Fig. 2 in section 3.1). Given that model mean trends reflect the response to external forcing due to the
averaging out of internal variability, this correspondence between the (multi-)model mean trend pattern and global mean trends on the one hand, and the first EOF and principal component on the other hand, implies that the first EOF can be assumed to reflect externally caused variance. The first principal component is therefore set to be the multi-model forced response best estimate. Each model's ensemble mean data is projected onto the first EOF of the multi-model mean to obtain model-specific forced responses, which form the targets for RR training.

Despite the high correlation between model global mean changes and EOF-projections, we prefer to use the EOF-based target as a default, since the first EOF captures the forced pattern of change, and its corresponding principal component time series captures the strength of that pattern. The first principal component is thus a reflection of the forced pattern strength (e.g. Marvel and Bonfils, 2013), meaning the forced response in all regions is somewhat reflected in this timeseries, and not averaged out as in the global mean. Yet, Fig. S10 shows that using model ensemble global mean as forced response target for detection does not lead to qualitatively different conclusions.

The procedure described above is visualised in a flowchart in supplementary Fig. S1, and the EOF patterns and model specific targets are shown in supplementary Fig. S2. The EOF-derived targets are relatively noisy, however, smoothing the forced response targets with a 21-year lowess filter before RR model training yields virtually identical results, indicating that the low frequency components of the targets govern the RR model configurations.

## 2.3 Ridge regression

The detection fingerprint is generated by regressing the forced response targets onto the spatiotemporal predictors using ridge regression, referred to as training. The resulting fingerprint is a spatial map of coefficients reflecting the relationship between predictors and forced response targets in model simulations. For the RR training procedure, we store the predictors in a 2D matrix $\mathbf{X}$ of size $n \times p$ (rows $\times$ columns), where each column corresponds to a gridcell in the coverage mask (with one extra column for the intercept), and the concatenated time series of three members per model make up the rows. The target variable is a vector $y$ of length $n$ consisting of a concatenation of the targets matching the predictors, i.e. the model member predictors predict their "own" model ensemble mean forced response target, to retain within-model physical consistency. The output of the RR training procedure is a coefficient vector $\beta$ of length $p$ such that:

$$y = \mathbf{X}\beta + \epsilon \tag{1}$$

Effectively, $\beta$ – the fingerprint – consists of coefficients for each gridcell in the coverage mask. Applying $\beta$ to model output ($\mathbf{X}_{mod}$) or observational data ($\mathbf{X}_{obs}$) then gives:

$$\hat{y}_{mod} = \mathbf{X}_{mod}\beta \tag{2}$$

$$\hat{y}_{obs} = \mathbf{X}_{obs}\beta \tag{3}$$

in which $\hat{y}_{mod}$ and $\hat{y}_{obs}$ are statistical predictions of the modelled and observed forced response; referred to as forced response estimates. In order to assess whether the external forcing has an effect that is distinct from internal variability, the fingerprint is applied to piControl model simulations to generate an unforced control forced response estimate; $\hat{y}_{mod,pi}$. All piControl data (4500 years) are input into the RR model to have a control distribution as large as possible.

$\beta$ is obtained numerically in R using the package glmnet for $k$-fold cross-validated ridge regression (Friedman et al., 2010; Simon et al., 2011). To determine $\beta$, the residual sum of squares plus the sum of squared coefficients (L2-norm) times a parameter $\lambda$ – the objective function – is minimised; see equation 4 for the minimisation problem.The regularisation parameter $\lambda$ can be tuned: the higher $\lambda$, the stronger the regularisation. This regularisation is the key characteristic of ridge regression.

$$\underset{\beta}{\mathrm{argmin}} \underbrace{(y - \mathbf{X}\beta)^T (y - \mathbf{X}\beta)}_{\text{Residual sum of squares}} + \lambda \underbrace{\beta^T \beta}_{\text{L2-norm}} \tag{4}$$

The RR cost function is minimised for a set of $\lambda$s through $k$-fold cross-validation, in which each fold contains data from one model. The simultaneous training and cross-validation on all models ensures that the resulting RR-fingerprint generalises well across models, reflecting where they agree and avoiding overfitting to any particular model. Training on many climate realisations ensures that the resulting RR-fingerprint leads to forced response estimates that are robust to internal variability. This means higher coefficients are given to gridcells where internal variability is smaller relative to the long-term trend, reflecting where signal-to-noise ratios and thus predictive value for the forced response estimates are higher. Figure 1 shows a visualisation of the ridge regression procedure, to intuitively clarify the relative roles of simulated data and observations in this approach. For a more detailed description of RR with glmnet, see (Friedman et al., 2010; Simon et al., 2011; Sippel et al., 2020).

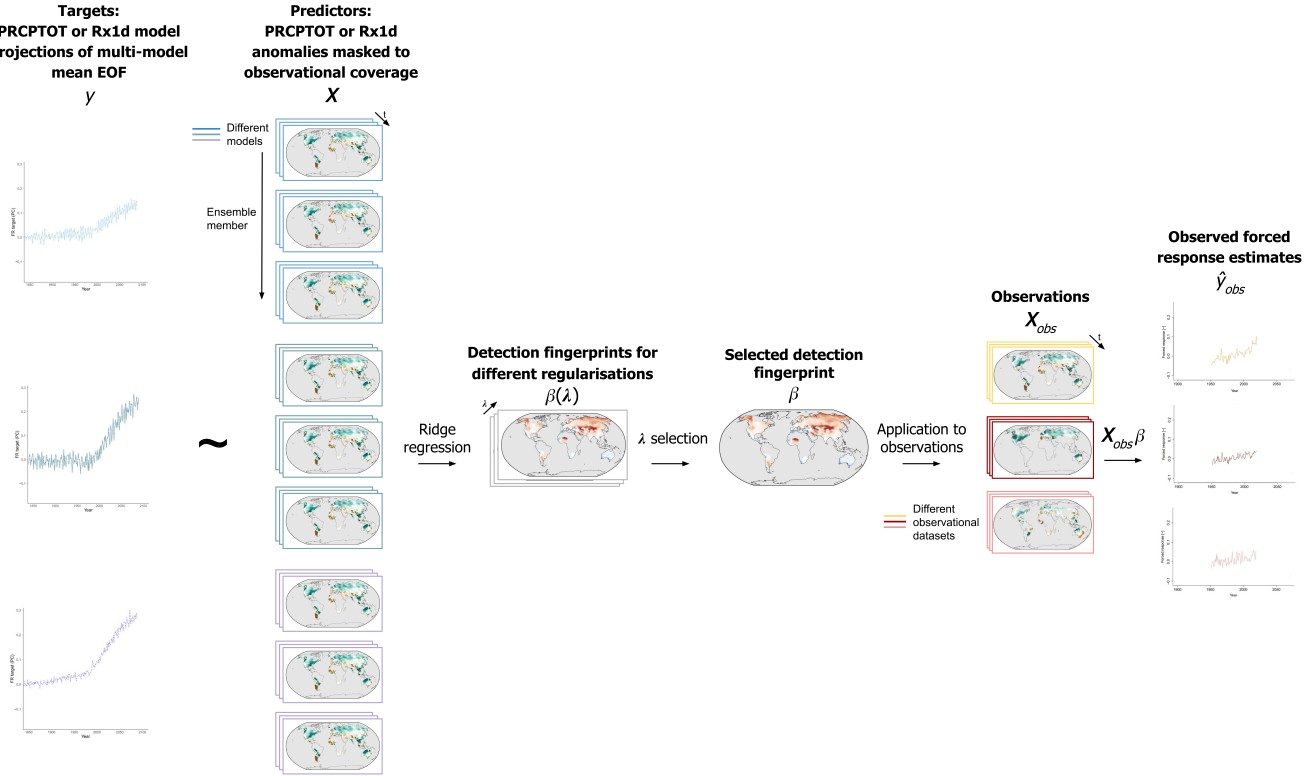

**Figure 1.** Flowchart of ridge regression model training procedure as well as application of detection model to observations.

Regularisation is used since the high number of predictors and their spatial dependence may otherwise lead to overfitting and unphysical coefficient fingerprints in which, for example, high positive coefficients are offset with adjacent negative co-

efficients. Regularisation acts to reduce overfitting, and results in a more homogeneous and smoother fingerprint, which more closely resembles the spatial scales of precipitation change patterns. It also increases generalisability of the fingerprint, and thus improves performance when applied to data that have not been seen in the training, such as observations.

There is no objective best approach to select the regularisation parameter $\lambda$. Smaller $\lambda$s reduce bias but increase variance (overfitting) whereas larger $\lambda$s do the opposite (Friedman et al., 2010; Simon et al., 2011). There are several common options for $\lambda$ selection, as default we use $\lambda_{sel}$, of which the definition and selection procedure are described in supplementary Sect. S1.3. We reason that the most regularised RR model with good performance is a good choice for the detection model, since model performance is very similar within the range of common $\lambda$s, whereas fingerprint interpretability decreases for $\lambda$s on the low end of the range. Sensitivities to $\lambda$ selection are addressed in Sect. 3.4.

Note that all the model simulations we use serve as input for RR training - i.e. the model forced response estimates shown in Sect. 3 are not out-of-sample application. Since the purpose of the RR model is to estimate the observed forced response from observational data that was not used in training, no independent model data sample for model forced response estimation is needed. Figure S4 in the supplement shows that pre-cross-validation results of the RR model applied to out-of-fold data are nearly identical to results of the final cross-validated model.

## 2.4 Forced trends and signal time of emergence

To assess the strength of the observed forced response estimates, we compare linear trends in $\hat{y}_{obs}$ for the different observational datasets to linear trends in the multi-model forced response best estimate $y$, as well as to the range of unforced trends given by the piControl forced response estimates $\hat{y}_{pi}$. In this study, we consider forced change to be detected if $\hat{y}_{obs}$ trend magnitudes lie outside the 95% range of trends from control simulations ($\hat{y}_{pi}$). The magnitude of the observed forced response estimate $\hat{y}_{obs}$ trends relative to the multi-model forced response best estimate $y$ trends indicates whether CMIP6 models over- or underestimate the real-world forced signals in PRCPTOT and Rx1d.

Besides linear trends, we also assess the signal-to-noise ratio (SNR) of the observed forced response estimates. We define SNR based on Hawkins et al. (2020), but we note that the signals here are global, as opposed to local signals in Hawkins et al. (2020). In order to separate the signal from the noise in forced response estimates, they are related to a smoothed, long-term forcing proxy as a covariate. Since global precipitation change scales with global temperature change, as described in the introduction, the long term trend in precipitation forced response estimates can be isolated using the long term trend in global temperature (see supplementary Fig. S5). Hence, the signal $S$ is defined as the observed forced response estimate regressed onto smoothed global mean surface temperature from Cowtan and Way (2014). Global mean surface temperature is smoothed with a 21-year lowess filter to remove interannual variability while keeping the long term trend (Hawkins et al., 2020). The noise $N$ is defined as the standard deviation ($\sigma$) of the residuals of this linear fit, i.e. $\sigma(\hat{y}_{obs} - S)$. The SNR ($\frac{S}{N}$) thus relates the observed forced response estimate signal to observed forced response estimate noise, providing a measure of signal emergence. The mean signal is centred to zero in the 21-year period 1951-1971, as we see minimal measurable forcing effects in precipitation metrics up to then. Between the 1951-1971 period and the present, the signal and thus the SNR increases. This

allows us to determine the time of emergence of a forced climate signal in mean and extreme precipitation, defined as the year after which the SNR consistently remains higher than 2.

## 3 Results and discussion

### 3.1 Precipitation change in model simulations and observations

In order to put the detection results in context, we first assess the general characteristics of historical precipitation trends in models and observations. Figure 2 shows maps of annual linear trends for PRCPTOT and Rx1d from the CMIP6 historical multi-model mean and the observational datasets over the 1951-2014 period.

For PRCPTOT (left panel) the model trend map (Fig. 2a) shows the well-established mean precipitation forced change spatial pattern, including a net global increase and an intensification of the global water cycle pattern (Douville et al., 2021). All observational datasets (figures 2c, e and g) contain features that resemble the multi-model mean forced patterns such as wettening in high latitudes. There are also some regions, such as Southeast Asia and West Africa, where observed and simulated trends have opposite signs. Uncertainties in the net precipitation response to the opposing forcing effects of greenhouse gases and aerosols in the second half of the 20th century, as well as internal variability, likely play a role in these discrepancies (Bonfils et al., 2020). The different observational datasets generally agree with one another, but a stronger trend over western North America in GHCNDEX (Fig. 2e) stands out. For HadEX3, GHCNDEX, and GPCC, models and observations agree on the sign of the PRCPTOT trend for 74%, 85%, and 68% of gridcells. For Rx1d (right panel) model trends (Fig. 2b) also reflect well-known changes, which are predominantly positive, especially over land. Observational records (figures 2d and 2f) agree in that they also feature mostly positive trends. For HadEX3 and GHCNDEX, models and observations agree on the sign of the Rx1d trend for 71% and 75% of gridcells. The fact that simulated trend patterns appear smoother and smaller in magnitude than observational trend patterns is primarily due to multi-model mean averaging.

The first EOFs underlying the RR targets look virtually identical to the simulated trend patterns – spatial correlation coefficients exceed 0.99 –, which implies that the first EOFs capture the forced trend signals (see supplementary Fig. S2).

### 3.2 Detection fingerprints and observed forced response estimates

Figure 3 shows the detection fingerprints, forced response estimates and forced trends for models and observations obtained with RR. The top panel of Fig. 3 shows the regression coefficient fingerprints that best predict the forced response while minimising variance due to internal variability and model disagreement, as described in Sect. 2. Only the fingerprint on the HadEX3-mask is shown here, GHCNDEX-masked and GPCC-masked fingerprints feature similar patterns where coverage overlaps, as shown in supplementary Fig. S6, increasing confidence in the generated RR-patterns and the method.

For both PRCPTOT (3a) and Rx1d (3b), large coefficients indicate changes with a high SNR and a time evolution that corresponds to the global forced response time evolution. Positive coefficients indicate changes with the same sign as the global forced response, whereas negative coefficients indicate changes with opposite sign. RR tends to rely on regions with

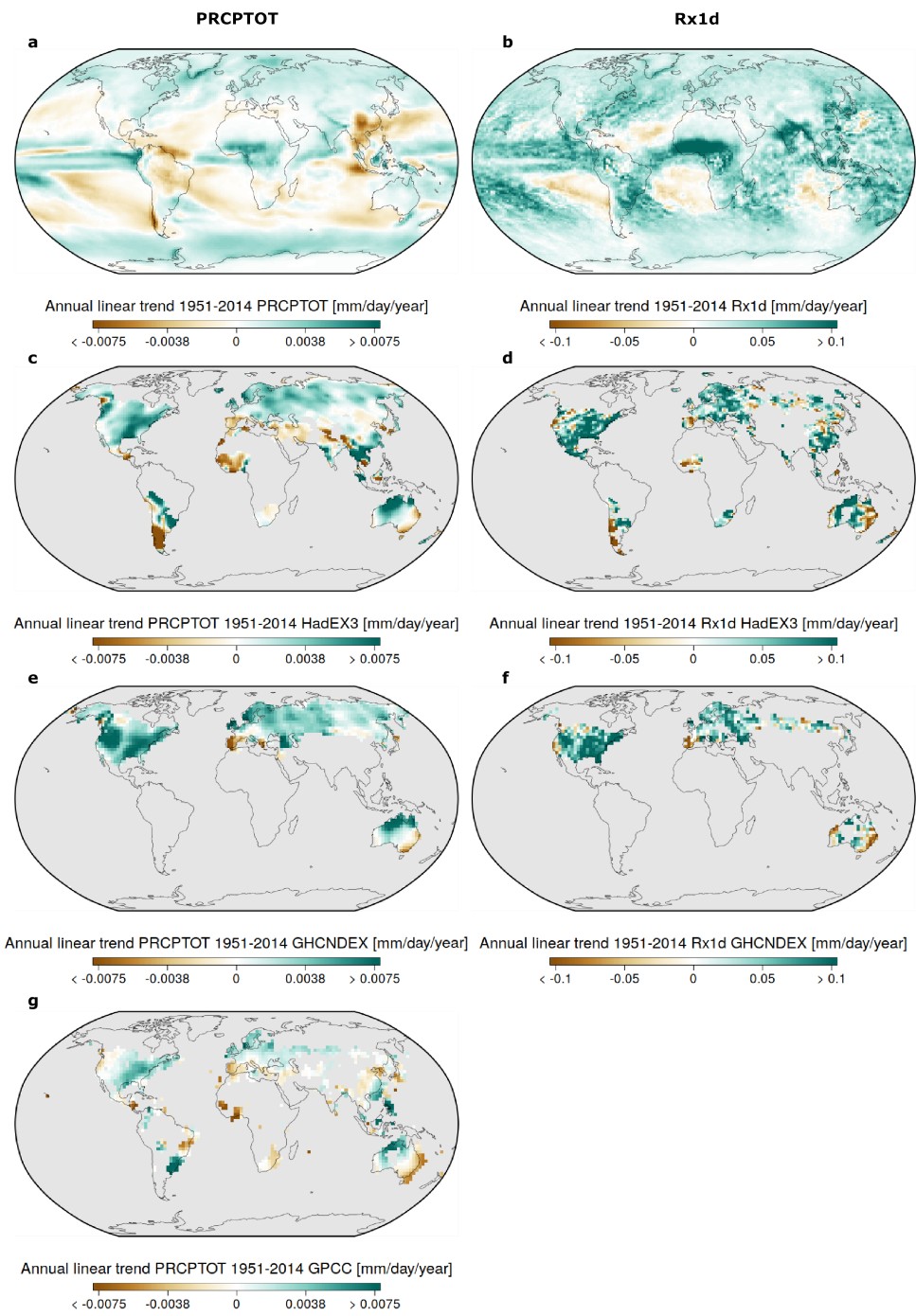

**Figure 2.** Mean total precipitation (PRCPTOT, left) and extreme precipitation (Rx1d, right) 1951-2014 annual linear trends in the CMIP6 multi-model mean (a, b); in HadEX3 observational data (c, d); in GHCNDEX observational data (e, f); and in GPCC observational data (g).

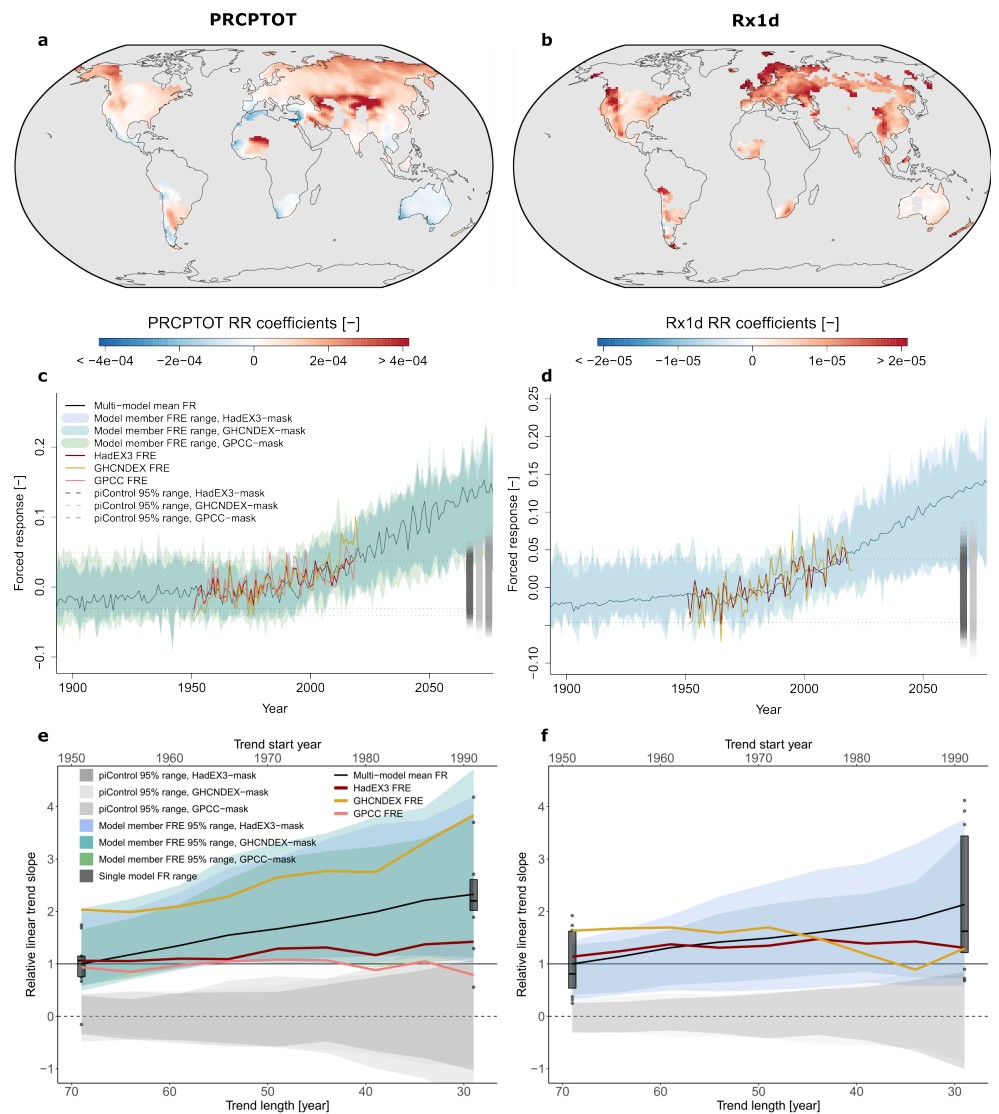

**Figure 3.** Annual mean total precipitation (PRCPTOT) and extreme precipitation (Rx1d) ridge regression detection fingerprints for HadEX3 coverage (top), corresponding forced response estimates (FRE) (middle), and linear forced response estimate trends as a function of trend period (bottom). In (c) and (d), black lines represent the multi-model forced response (FR) best estimate, coloured shading the full range of model forced response estimates, coloured lines the observed forced response estimate, grey dashed lines the unforced internal variability of the forced response pattern (piControl forced response estimate 95% range (2.5% to 97.5% quantile range). In (e) and (f), black lines represent the multi-model forced response best estimate trend, coloured shading the 95% range of model forced response estimate trends, coloured lines the observed forced response estimate trends, grey shading the piControl trends, black boxplots the model ensemble mean target trends. Trends are computed from a variable start year until the end year of the observational time series (2018 for HadEX3, 2019 for GPCC, 2020 for GHCNDEX, 2018 for multi-model forced response best estimate and model ensemble mean trend). Start years vary between 1951 and 1991 with increments of 5. piControl trends are computed over periods equally long as the corresponding forced trends. All linear trends are normalised with respect to the multi-model forced response best estimate 1951-2018 trend, i.e. the leftmost point on the black line.

smaller but more robust changes for forced response prediction, whereas some regions with larger changes such as the tropics contribute less to the prediction, due to their high internal variability and uncertainty (Kent et al., 2015). Regions with very small coefficients coincide with where CMIP6 models have been shown to disagree strongly on the sign of change, for example the location of the transition from negative (south) to positive (north) coefficients in Europe for PRCPTOT, and central America and the whole of Australia for both PRCPTOT and Rx1d (Douville et al., 2021; Giorgi et al., 2014; Westra et al., 2013; Sun et al., 2022; Kotz et al., 2022; Kent et al., 2015). For many of these regions, the disagreement among models about precipitation change can be traced back to the influence of circulation changes on precipitation changes, which are particularly uncertain; for example, expansion of subtropical dry zones.

Several specific features that reflect the forced response pattern of PRCPTOT and Rx1d can be distinguished in the finger-prints. The PRCPTOT fingerprint features negative coefficients in southern Europe and northern Africa, as well as South-Africa and Australia, which reflect the drying pattern corresponding to expected forced change in the hydrological cycle (Douville et al., 2021). Additionally, the climatologically wet Pacific Northwest exhibits positive coefficients as the air that rains out due to orographic lift by the Cascade, Coastal, and Olympic mountain ranges becomes increasingly moist with climate change. The rain shadow on the lee side features negative coefficients. The Rx1d fingerprint looks more homogeneously positive than for PRCPTOT, reflecting the expectation of a positive trend in Rx1d over almost all land regions, as seen in Fig. 2 (Pfahl et al., 2017). The strong positive coefficients in Northern Europe and the North-American West Coast can likely be explained by the systematic nature of extreme precipitation in these regions – wet ocean westerlies making landfall –, which results in a consistent response to increased atmospheric moisture and thus high predictive value for the global forced response (Pfahl and Wernli, 2012). The smaller positive or even negative coefficients in the tips of South America and South-Africa correspond to regions where dynamical changes are known to mask thermodynamic increases in Rx1d (Pfahl et al., 2017; Kotz et al., 2022; Li et al., 2021).

The similarities in the maps for PRCPTOT and Rx1d indicate that the signs of change in PRCPTOT and Rx1d correspond in most regions, pointing towards a precipitation distribution shift to higher mean and extreme precipitation levels. As mentioned, however, the magnitude of the increase is larger for Rx1d than for PRCPTOT, and regions where negative changes in PRCPTOT exist in combination with positive changes in Rx1d are also found. This corresponds to widening of the precipitation distribution and complies to the expected forced increase in precipitation variability (Zittis et al., 2021; Pendergrass et al., 2017). From an impacts perspective, this could imply that the background climate in some regions dries while wet extremes become more intense, which can increase both drought and flood risks (Tramblay et al., 2019).

The middle panels of Fig. 3 show the forced response estimates for PRCPTOT (3c) and Rx1d (3d), which are the result of applying the RR-fingerprints to model simulations and observational data. The green/blue shading shows the range of forced response estimates from CMIP6 individual member data for all observational masks. The consistency of the trend in the model forced response estimate envelopes and the multi-model forced response best estimate (black line) confirms that the RR-fingerprints are indeed suited to capture the global climate change signal in PRCPTOT and Rx1d from spatially incomplete model data. The model forced response estimates show a slight high bias in early years where the target is at the low end of

its range, and a slight low bias in late years where the target is at the high end of its range. This effect is expected since the regularisation "trades" some goodness of fit for generalisability, and makes the forced response estimates more conservative.

The coloured lines show forced response estimates from observations. The observed forced response estimates lie well within the model forced response estimate range, exhibit similar variance, and follow the trend of the multi-model forced response best estimate. These trends in the observed PRCPTOT and Rx1d forced response estimates indicate that the strength of the forcing pattern increases in observations indeed, and generally agrees with model projections.

The grey dashed lines show the 95% range of the forced response estimate from unforced piControl data, and the piControl forced response estimate distribution is also shown as a point cloud to the right of the timeseries, reflecting the internal variability range of the detection pattern. Over the historical period, observed forced response estimates have moved from the middle towards the upper bound of the piControl range, and the multi-model forced response best estimate and model forced response estimates leave the piControl range still in the first half of the 21st century. All of the above points to the unambiguous detection of forced climate change in annual PRCPTOT and Rx1d in all observational datasets used.

We note that the GHCNDEX forced response estimate for PRCPTOT shows a distinct uptick towards the end of the record, which likely is related to the coverage of GHCNDEX being almost exclusively in the higher Northern latitudes (more so than for the other datasets), which contribute disproportionately in these particular years. However, based on the analysis here, we cannot differentiate whether this is an artefact, internal variability, or indicative of an increasing forced rate of change in PRCPTOT.

Besides visual inspection of forced response estimate timeseries, quantitative detection statements can be made based on the trends in these timeseries. For lack of evidence for a particular forced response polynomial, the high amount of noise in observed forced response estimates, and ease of interpretation, linear trends are assumed. The trends in PRCPTOT and Rx1d, however, are not constant with time in the period of interest, so we also include the dependence of forced trend estimates on the length and start year of the trend period. Figures 3e and 3f show a quantitative overview of the linear trends of targets and forced response estimates as a function of start year and trend length. Findings are normalised with respect to the 1951-2018 multi-model forced response best estimate trend.

Since forced trends in both PRCPTOT and Rx1d only begin to appear around 1975, the multi-model forced response best estimate trends are larger in more recent trend periods that omit earlier years (toward the right side of the x-axis). The model forced response estimate trend 95% ranges (green/blue shading) are reasonably symmetric around the multi-model forced response best estimate and include the majority of the ensemble mean target trends (black boxplots). This agreement of model forced response estimate trends with forced response target trends shows that the RR-model does well in estimating the forced trend magnitudes. Part of the intermodel spread in both forced response targets (boxplots) and model forced response estimates is explained by the different climatological levels of precipitation among models and their different climate sensitivities, and in part by model uncertainties in temperature-independent precipitation adjustments (Fläschner et al., 2016). Despite the large spread, there is only little overlap of the model forced response estimate trend range and the piControl trend range (grey shading), implying that almost the entire range of model forced response estimate trends lies outside the range of trends possible in an unforced climate. This confirms once again that there is a robust forced signal in mean and extreme precipitation.

The observed forced response estimate trends for both PRCPTOT and Rx1d (coloured lines) exceed zero, lie within the model forced response estimate trend range and outside the piControl range for all trend lengths (apart from GPCC trends over the most recent 40 years or shorter), confirming detection of forced change in observations. Although forced change is unambiguously detected in all datasets, the degree of observed change depends on the observational dataset considered.

GHCNDEX yields higher observed forced response estimate trends than the multi-model forced response best estimate trend for PCRPTOT whereas HadEX3 and GPCC yield lower observed forced trends. Hence, GHCNDEX suggests that CMIP6 models *underestimate* the forced change in PRCPTOT, whereas HadEX3 and GPCC suggest CMIP6 models *overestimate* it. The higher trends in GHCNDEX PRCPTOT are partly caused by the few high outliers towards the end of the GHCNDEX timeseries mentioned earlier, but also persist when these are removed from the timeseries.

For Rx1d, GHCNDEX and HadEX3 forced response estimates show more similar trends, although GHCNDEX trends again exceed HadEX3 trends for trends that include years before 1975. For more recent periods, GHCNDEX shows smaller trends than HadEX3. In general, forced response estimate trend increases in both observational datasets flatten out for periods from 1975 to the present, where trends smaller than the multi-model forced response best estimate are found. By contrast, periods including years prior to 1975 suggest observed trends larger than in the multi-model forced response best estimate.

Whereas the magnitudes of the observed forced response estimate trends differ considerably among the observational datasets in some cases, as highlighted above, the relative trend fluctuations over time resemble each other in all datasets. This consistency of results increases confidence in the robustness of the method, and suggests that differences in spatial coverage and data operations among observational datasets are the main sources of uncertainty in observed forced trend estimation.

Although these results are sufficient to conclude that the detection of forced change in global mean and extreme precipitation is unequivocal, internal variability and short record length preclude our ability to conclude whether the observed change is weaker or stronger than models suggest. The use of multiple observational datasets and the time-dependent view of the forced trends in observations shows that the magnitude of forced change detected in precipitation observations is sensitive to choices on the specifics of the analysis. In previous studies, opposing conclusions have been drawn as to the magnitude of forced precipitation change in observations relative to model simulations, as noted in the introduction. Our results show that both conclusions can be true, depending on the observational dataset and the forced trend metric used.

Confidence in these results is strengthened by the consistency of the variability in the model forced response estimates and observed forced response estimates. The residuals of the linear fit to the observed and modelled forced response estimates have comparable distributions, shown in supplementary Fig. S7. This residual variance consistency justifies the use of the model-derived RR-fingerprint on observations, and decreases the likelihood of spurious detection. Confidence in the method is also enhanced by its robustness to target metric; the results above also hold when the global mean is used as forced response target, as shown in supplementary Fig. S10. In addition, the main finding that the magnitude of detected forced changes in precipitation observations relative to simulated forced changes depends on the dataset, holds also when relative precipitation metrics such as percentage change or percentage change per temperature change are assessed, shown in supplementary Sect. S2.3.

### 3.3 Detection based on relative spatial patterns of precipitation alone

It is not surprising that forced change in mean and extreme precipitation can be detected on a global scale, given the consensus on global mean increases in PRCPTOT and Rx1d with increasing global temperatures. A more powerful detection statement can potentially be made, however, if forced change can be detected in the spatial pattern of precipitation observations alone, excluding the global mean trend. Therefore we attempt to construct RR-models based on training data from which the global mean trend is removed (detrended) by subtracting the coverage-masked, area-weighted spatial mean for each time step. The resulting fingerprints are then applied to observations which are detrended in the same way.

Figure 4 (top) shows the RR-fingerprints for detrended predictors on the HadEX3 mask for mean and extreme precipitation (GHCNDEX and GPCC masked fingerprints look similar where coverage overlaps, see supplementary Fig. S8). In these fingerprints, negative coefficients indicate a change that is in phase with the forced response but of opposite sign (inversely correlated), which can point to a decrease, but also to an increase with a smaller slope than the (coverage-masked) global mean increase. The latter is the case when coefficients flip sign from positive in Fig. 3, where the trend is included, to negative in Fig. 4. Positive coefficients in Fig. 4, on the other hand, indicate increases with slopes larger than the global mean increase. As for the fingerprints in Fig. 3, large coefficient magnitudes in Fig. 4 signify high SNR but not necessarily large changes in an absolute sense.

Inspection of the detrended fingerprints leads to several interesting insights. Both for PRCPTOT and Rx1d we see that some regions with large regression coefficients flip sign. As stated above, this reflects high-SNR changes of the same sign but with a smaller rate of change than the global mean. This again shows that RR relies to a large degree on regions with small but consistent changes, for example the Tibetan plateau for PRCPTOT. This effect is even more strongly visible in the Rx1d fingerprint, since continental Europe and western North-America – the regions with strong positive coefficients in the fingerprint with global mean included – flip sign. Rx1d increases in these regions are thus smaller than global mean, yet strong indicators of forced change in the context of internal variability and uncertainty among models. Rx1d increases likely are dominated by large tropical rainfall increases, as indicated by persistent positive coefficients in tropical regions and the North-American and Asian east coasts.

For both PRCPTOT and Rx1d, the regions of primary importance largely remain the same between the two fingerprint types. This implies that the fingerprint with the global mean trend included picks up on high SNR forced signals beyond the large scale mean increase.

The observed forced response estimates for detrended PRCPTOT and Rx1d (coloured lines) in figures 4c and 4d show a clear positive trend that is in agreement with the multi-model forced response best estimate (black line). Recall that the forced response estimates in figures 4c and 4d are derived from observations from which the global mean is removed, meaning that the relative spatial patterns of PRCPTOT and Rx1d alone exhibit a clearly detectable forced long term trend. However, the larger spread in the observed forced response estimates compared to figures 3c and 3d shows that detrending of the predictors – i.e. removing part of the signal – results in larger variability of the observed forced response estimates, which reduces the SNR ratio of the trends.

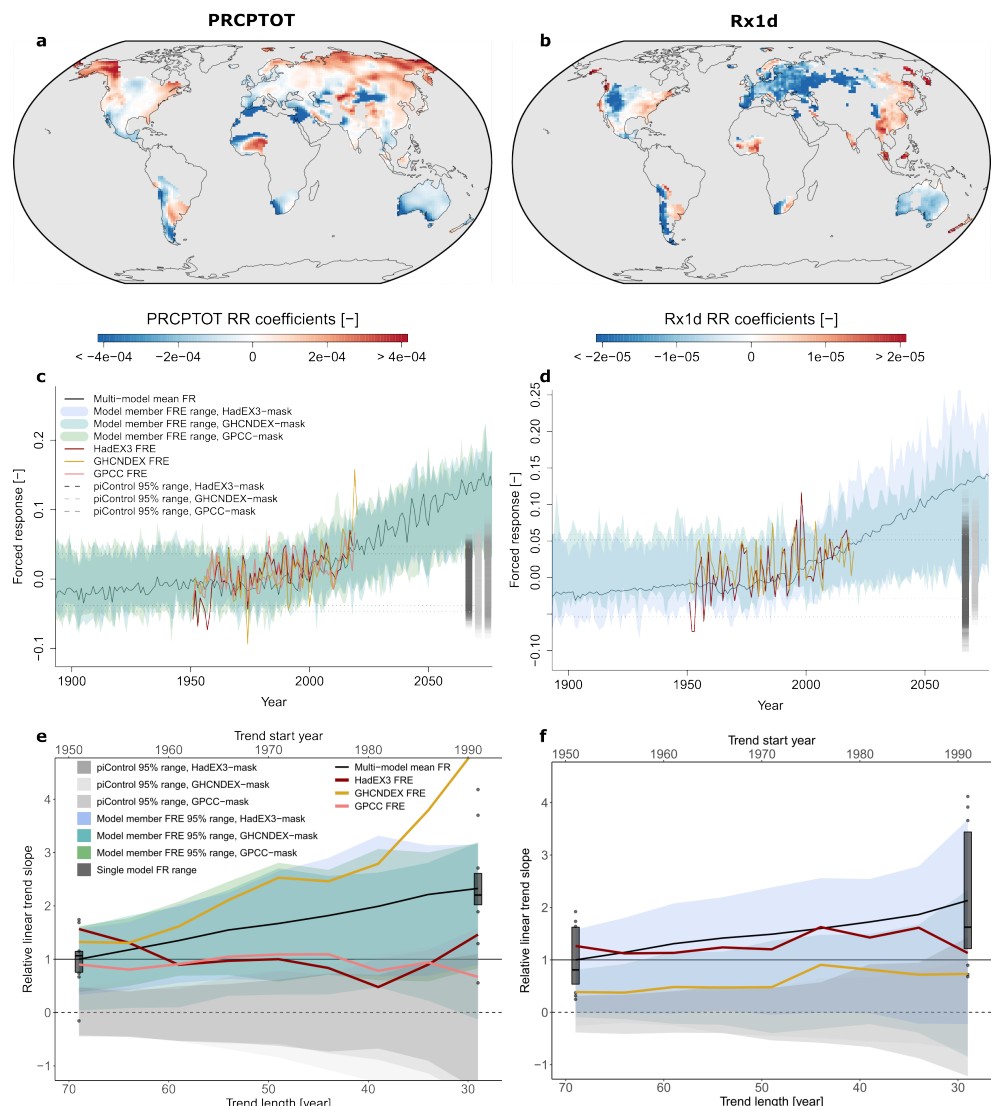

**Figure 4.** As Fig. 3 but for detrended predictors. Detrending implies removing the masked, area-weighted spatial mean from the model member data and observational data for each time step.

The model forced response estimates also exhibit larger variability, which causes the forced and piControl trend ranges (green/blue and beige/gray) to overlap more (figures 4e and 4f), indicating a lower detectability of forced change in detrended model data. Particularly for GHCNDEX-masked Rx1d data (green shading in Fig. 4f), the low coverage in combination with detrending removes so much information that forced response estimation from model data is substantially impaired. With weaker regularisation the forced trend still cannot be estimated from detrended GHCNDEX data.

Despite the reduced information given to the RR-model in the detrended case, figures 4e and 4f show that forced change is still detected using the spatial pattern alone. The observed forced trends lie outside the piControl range and are in reasonable agreement with the multi-model forced response best estimate trends for longer trend lengths. For shorter trend lengths, the higher variability in the forced response estimates leads to higher trend variability as well. Consistent with figures 3e and 3f, we see that HadEX3 and GPCC show smaller PRCPTOT trends than the multi-model forced response best estimate, whereas GHCNDEX shows larger trends. We note that very high GHCNDEX PRCPTOT forced response estimate trends seen here are untrustworthy given that GHCNDEX residual consistency in the detrended setup is insufficient (see supplementary Fig. S9). HadEX3 Rx1d trends agree very well with the multi-model forced response best estimate trends.

A possible interpretation of the forced change detection in HadEX3 Rx1d but lack thereof in GHCNDEX Rx1d, is that the forced response in Rx1d can be detected in absence of the global mean, but that sufficient coverage is necessary. The seemingly higher sensitivity of Rx1d to detrending is likely because the Rx1d forced response is more spatially homogeneous, which implies that global mean detrending removes much more of the signal than for the spatially heterogeneous PRCPTOT forced response. Taken together, the above shows, first, detection of forced change in mean and extreme precipitation beyond a global mean trend, and second, the power of RR for signal extraction from high-dimensional noisy data. Finally, the fact that the relationship between relative spatial precipitation patterns and the forced precipitation trend derived from climate model simulations (the ridge model) holds in observations, suggests accuracy of the CMIP6 climate models in simulating the processes relevant to the spatial pattern of forced change in mean and extreme precipitation.

### 3.4 Time of emergence

The forced response estimates and trends in figures 3 and 4 provide evidence that the observed forced trends are larger than the unforced piControl trend distribution, both with and without global mean signal. Figure 5 provides the SNR as a quantitative assessment of the observed forced response estimate signal strength relative to the observed forced response estimate variability, as defined in Sect. 2.4. Besides the default case (solid lines, corresponding to the forced response estimates in figures 3c and 3d), the detrended SNR (dotted, corresponding to the forced response estimates in figures 4c and 4d) as well as the SNR for less regularised RR-models with minimal cross-validated mean squared error ($\lambda_{min}$) (dashed) are shown (Friedman et al., 2010; Simon et al., 2011). SI Sect. S2.4 shows the sensitivity of time of emergence to additional method choices.

Time of emergence – the time after which the SNR consistently lies above 2 – is indicated by the vertical lines. We consider the above definition of time of emergence a consistent measure of effective SNR in the real climate, since both signal and noise are derived from observations. To assess the effects of possible autocorrelation within observational residuals as well as possible biases due to the relatively small sample size, we also compute SNRs with respect to a noise measure derived from forced response estimates of control simulations, as in Hawkins and Sutton (2012). This definition of SNR results in similar outcomes (not shown).

Overall, figures 5a and 5b show emergence of forced change within four years of 2000 in both PRCPTOT and Rx1d in GHCNDEX and HadEX3 for the default setup (solid lines). The nearly identical time of emergence for PRCPTOT and Rx1d obtained using our method of forced response estimation is noteworthy, given earlier suggestions of a later emergence of

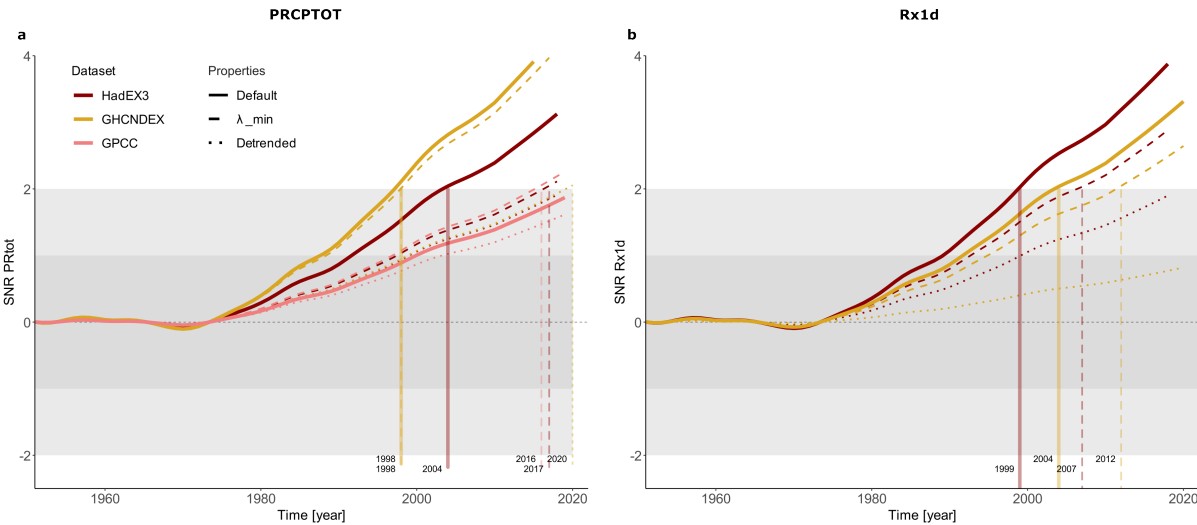

**Figure 5.** SNRs of mean total precipitation (PRCPTOT) (a) and extreme precipitation (Rx1d) (b) forced response estimates in GHCNDEX, HadEX3 and GPCC, including senstivities to regularisation parameter (dashed) and global mean detrending (dotted). Exceedance of an SNR of 2 implies emergence. Signal is defined as forced response estimate regressed onto 21-year lowess filtered global mean surface temperature, noise as residuals of this fit.

PRCPTOT due to higher uncertainties and internal variability (Fischer et al., 2014; Fischer and Knutti, 2014). The exact
time of emergence differs between datasets, as expected given the different trends seen in Fig. 3, and GPCC PRCPTOT does
not show emergence at all due to its weaker trend combined with high variability. We note that GPCC is constructed using
a different gridding procedure than HadEX3 and GHCNDEX, and also our handling of GPCC is different due to the need
to specify a coverage mask based on station density, whereas HadEX3 and GHCNDEX provide their own coverage masks.
Interestingly, Rx1d time of emergence is earlier in HadEX3, despite larger long-term linear trends in GHCNDEX. This reflects
the higher efficiency of RR in reducing variance while capturing the signal for the higher spatial coverage of HadEX3.

    The benefit of regularisation becomes evident when comparing the default case to the $\lambda_{min}$ setup, where regularisation is
such that the training cross-validation error is smallest (see supplementary Sect. S1.3 for a more extensive definition). For
$\lambda_{min}$ forced response estimates, SNR is lower and time of emergence is later (despite slightly larger forced response estimate
trends), due to the increased variance in the $\lambda_{min}$ forced response estimates caused by overfitting on the training data. For
GPCC this effect is not seen, since the regularisation for GPCC in the default case is weak, i.e. $\lambda_{sel}$ and $\lambda_{min}$ are not far apart
and variance in the forced response estimate hardly increases for $\lambda_{min}$.

    Lastly, the effect of removing the global mean from the data (detrended), discussed in the previous section, is shown in the
dotted lines. Due to the increased variance in the forced response estimates, SNRs decrease and times of emergence increase.
Yet, signals have emerged or are close to emergence in all detrended cases except for GHCNDEX Rx1d, once again confirming
the detection of forced climate change in spatial patterns of PRCPTOT and Rx1d.

All of the above points to detection and emergence of a forced response in observations of mean and extreme precipitation, robustness of the detection method, and representation accuracy of forced and internal variability patterns of precipitation in climate models.

## 4 Conclusions and outlook

We demonstrated the detection and emergence of forced change in mean and extreme precipitation beyond internal variability using a recently-introduced detection method based on regularised linear regression. We generate regression models for detection of forced change based on climate simulations, consisting of physically interpretable fingerprints that optimise the signal-to-noise ratio. We detect forced trends in both mean and extreme precipitation that lie outside the piControl range of unforced variability in three different observational datasets. The unequivocalness of the detection of forced change is further

demonstrated by the sustained detection from the spatial pattern of precipitation alone, after subtracting the global mean trend from the data. A similar result was shown earlier for mean precipitation (Barnes et al., 2019), and is extended here to extreme precipitation. This finding also reinforces confidence in the ability of CMIP6 models to respresent processes that govern the (large-scale) spatial distribution of precipitation. Simultaneous emergence of the forced signal from internal variability in both PRCPTOT and Rx1d demonstrates the value of RR-based fingerprint construction for high signal-to-noise ratio estimation of

forced responses.

Despite the robustness of the results, the relative magnitude of forced trends in observations and models depends on the period over which trends are calculated, as well as on the observational dataset. We show in supplementary Sect. S2.3 that the dependency of trend magnitudes on trend period and dataset remains when we use different metrics of precipitation, such as percentage change per degree of warming. These sensitivities emphasise the difficulty associated with quantitative assessment

of observed changes in precipitation, as demonstrated by apparent contradictions in recent studies on whether models under- or overestimate the observed changes. Figure S14 in the supplement contains an overview of D&A studies on mean and extreme precipitation, showing the lack of consensus on observed forced trend strength across studies. This overview reveals that, in line with our study, model and observational uncertainties, changing observation station densities, internal variability, and structural differences between model simulations and observational data lead to different results, even when similar time

periods and precipitation metrics are considered. Against the backdrop of such uncertainty, further development of methods such as ours, that optimise for high SNR and support intuitive physical interpretation of results can be of great value.

It is important to note that the influence of Northern Hemisphere (NH) precipitation is disproportionately strong in this analysis. Part of this larger NH contribution may be due to stronger or earlier emergence of a forced response, which has been found in other studies (King et al., 2015). However, the uneven distribution of measurement stations over the global land plays

a large role as well. Therefore, the global detection found in this study may not be representative for smaller sub-regions, especially outside of the NH. Furthermore, preliminary results suggest that detection is sensitive to seasonal process specifics – we find that forced change is not detected in June-July-August (NH summer), on both global and NH specific scales (see supplementary Fig. S21). This is potentially related to the convective nature of precipitation in NH summer. We provide a

preliminary application of the method to regional and seasonal scales in supplementary Sect. S3. Extending the approach to
D&A of precipitation changes on regional and seasonal spatiotemporal timescales is of great importance to increase practical
relevance of the results for risk assessment and adaptation.

In this study we do not explicitly separate the effects of different forcings (GHG, aerosols, natural). We assume, however,
that the analysis primarily pertains to GHG-forcing since the RR fingerprint is based on historical-SSP245 projections through
2100, when GHG forcing dominates (Chen et al., 2021). Nonetheless, an extension of the present study explicitly separating
different forcings would be insightful to further characterise the effects of different forcing agents in the real climate, and
potentially identify sources of disagreement between models and observations. This is important as Wu et al. (2013) shows
that different models may agree on the simulated response to all forcings combined, while they differ greatly on separate
responses to GHG and aerosol forcings alone. Correct simulation of the relative effects of different forcing agents is important
for scenario development and climate action targets, meaning further investigation of these model discrepancies is imperative.
RR-based analyses may enable establishment of observational constraints on the precipitation response to different drivers,
which can help constrain projections of near term changes in mean and extreme precipitation.

*Code and data availability.* All original CMIP6 data used in this study are publicly available on https://esgf-node.llnl.gov/projects/cmip6/.
HadEX3 and GHCNDEX data are publicly available on https://www.climdex.org/access/. GPCC data are publicly available on https://
opendata.dwd.de/climate_environment/GPCC/html/gpcc_normals_v2020_doi_download.html. Preprocessed data and ridge regression model
training code are available on [to be added], additional code is available upon request.

*Author contributions.* IEdV: conceptualisation, methodology, software, formal analysis, writing, visualisation. SS: conceptualisation, method-
ology, software, writing - review & editing, supervision, funding aqcuisition. AGP: conceptualisation, writing - review & editing, supervision,
funding aqcuisition. RK: conceptualisation, writing - review & editing, supervision, funding aqcuisition.

*Competing interests.* The authors declare no competing interests.

*Acknowledgements.* We would like to thank N. Meinshausen, E. Fischer, J. Zeder and M. Egli for helpful and stimulating discussions. We
thank U. Beyerle, R. Lorenz, and L. Brunner for the preparation and maintenance of CMIP6 data. We acknowledge the World Climate
Research Programme's Working Group on Coupled Modelling, which is responsible for CMIP, and we thank the climate modeling groups
for producing and making available the model output. For CMIP, the U.S. Department of Energy's Program for Climate Model Diagnosis
and Intercomparison provides coordinating support and led development of software infrastructure in partnership with the Global Organiza-
tion for Earth System Science Portals.We acknowledge the Climdex website www.climdex.org for making available observational climate
indices used in this study. IEdV and SS acknowledge funding received from the Swiss National Science Foundation within the project

"Combining theory with Big Data? The case of uncertainty in prediction of trends in extreme weather and impacts" (grant no. 167215). SS acknowledges funding from the Swiss Data Science Centre within the project "Data Science-informed attribution of changes in the Hydrological cycle" (DASH; C17-01) and within the European Union H2020 project "Artificial intelligence for detection and attribution" (XAIDA; grant no. 101003469). AGP was supported by the U.S. Department of Energy, Office of Science, Office of Biological & Environmental Research (BER), Regional and Global Model Analysis (RGMA) component of the Earth and Environmental System Modeling Program under Award Number DE-SC0022070 and National Science Foundation (NSF) IA 1947282, and by the National Center for Atmospheric Research (NCAR), which is a major facility sponsored by the NSF under Cooperative Agreement No. 1852977.

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
