# Peer review of "Robust global detection of forced changes in mean and extreme precipitation despite observational disagreement on the magnitude of change"

_EGUsphere, 2022_

## Author Comment (AC1)

Comments of Referee #1
Response to Referee #1

This paper proposes a ridge regression approach to the detection and attribution of externally forced changes in mean and extreme precipitation. This is an interesting idea that certainly merits exploration, but before devoting a lot of time to understanding the details of the paper and the results that are obtained, I think it is necessary for the authors to better explain their method and to situate it within the pantheon of methods that are already available for detection and attribution.

We would like to thank the reviewer for this important remark. We fully agree that additions to the already densely populated D&A field require justification. In the following more specific comments, we hope to address this, and in a revised paper we will make sure that the method is properly explained along with its positioning and advantages/disadvantages within the broader field of D&A methods. This method adopts several aspects of traditional detection methods (e.g. Santer et al., 2013, 2018; Marvel et al., 2013, 2020; Bonfils et al. 2021; connection explained below in detail), but is indeed not directly following from or equivalent to "optimal fingerprinting" - the method referred to by the referee below. However, we also note that our paper does not intend to propose a new method; different applications of the method we use have been described and used in several papers in recent years, e.g. Sippel et al. (2020).

Ridge regression is a technique that "regularizes" regression problems, such as that described in equation (1) of the paper, in which the predictor variables contained in matrix X are multicollinear. In the generalized least squares formulation of the regression used in detection and attribution this matrix is composed of model simulated estimates of the responses to external forcing in the form of space-time patterns of change. Depending on variable, period considered, domain of interest and how data are processed, the expected space-time patterns of responses to different forcing factors (often called fingerprints) can be strongly correlated, which results in a regression "design matrix" X that may be ill conditioned. Ridge regression is a technique that can be used to overcome this problem, although I imagine at the cost of introducing some bias into the estimated signal scaling coefficients β. Note that referring to these coefficients as "fingerprints" seems unusual to me.

The concept of regularization, however, also arises in a second way in the detection and attribution problem. Considering again equation (1), the generalized least squares approach (and also its total least squares extension) requires knowledge of the variance covariance matrix of the residuals ε, which are regarded as resulting from natural internal climate variability. Thus, the variance-covariance matrix is generally estimated from unforced control simulations, using as many climate-model simulated realisations of ε as possible. Even though many climate-model simulated realizations of ε are now generally available, the estimated variance-covariance matrix may not be of full rank or may remain uncertain. Thus, it is also often regularized, using an approach similar to the regularization used in ridge regression, but applied to the noise term rather than the signal term of equation (1). See Ribes et al (2013a, doi:10.1007/s00382-013-1735-7, and 2013b, doi:10.1007/s00382-013-1736-6). Presumably one would want to regularize both aspects

of the problem, and also take signal uncertainty into account as is done in the total least squares approach to the regression problem (see again Ribes et al., 2013a and 2013b, and also Allen and Stott, 2003, doi:10.1007/s00382-003-0313-9).

How the combined model represented by equations (1-3) relates to existing techniques, and now the noise that results from internal variability comes into play and is accounted for in their subsequent application in the paper is not made clear, and I think should be clarified before results can be considered.

Thank you for your comment, and we agree that it is crucial to clarify the relationship to other methods in the D&A spectrum. Thanks also for the very nice description of the motivation behind the optimal fingerprinting method that addresses the multicollinearity problem of spatiotemporal data used in D&A, and the estimation of the variance-covariance matrix of internal variability.

Please note, however, that there is one important difference between the method we employ in our paper, and optimal fingerprinting (e.g., Allen & Stott, 2003; Ribes et al., 2013a, 2013b): In optimal fingerprinting, the observations (in space/time) are regressed onto the model's (space/time) simulated response patterns, using an estimate of the variance-covariance matrix of internal variability; and inference is made via the signal scaling factors of the regression. Unlike optimal fingerprinting, however, our method does not regress observations onto the model's simulated space-time response patterns. Instead, the goal of our regression step (Eq. 1) is to derive a projection of the model's output (i.e. space/time data of PRCPTOT and Rx1d) onto a one-dimensional detection space. We achieve this via (ridge) regression of each model's (one-dimensional) global forced response estimate Y (obtained by standard techniques) of length $n$ (given by the number of CMIP6 model simulations and years considered) onto the models' simulated Rx1d or PRCPTOT patterns X (including internal variability) at $p$ grid cells (where $p$ is given by observational coverage). This means we obtain a set of regression coefficients $\beta$ of length $p$, which represents a spatial pattern that best maps each individual modeled Rx1d/PRCPTOT pattern onto its forced response estimate Ŷ. We call this $\beta$ the fingerprint. In a second step we use the fingerprint $\beta$ to map observations (and piControl simulations) into the one-dimensional space in which detection is assessed. In our paper, we compare the trends in simulated and observed forced response estimates, but we do not formally determine any analogues to scaling factors in optimal fingerprinting. As correctly pointed out by the reviewer, our $\beta$ coefficients (fingerprint) are thus fundamentally different to the signal scaling factor central to optimal fingerprinting, which is sometimes also called $\beta$.

We would also like to clarify the role of regularisation by juxtaposing the use of regularisation in our method with the use of regularisation in optimal fingerprinting studies (e.g. Ribes et al., 2013a, 2013b). In our application, regularisation aims to reduce the effects of internal variability on the forced response prediction, i.e., regularisation is the way we account for the noise resulting from internal variability - one of the concerns raised by the referee. To make this more concrete: if we would use ordinary least squares, the regression step (Eq. 1) would result in a highly overfit map of coefficients $\beta$ that e.g. employs high weights on low-amplitude anticorrelated grid cells to explain a minor part of the variance, due to overdetermination of the problem. Therefore, we apply ridge regularisation, which penalises the squared magnitude of coefficients (L2-norm) in a way that leads to lower, spatially more

uniform, but non-zero coefficients. In this way, variance in the forced response estimate due to internal variability is reduced, i.e. the robustness to internal variability is increased and the $\beta$ fingerprint is more generalisable across models and between models and observations. As outlined by the referee, optimal fingerprinting employs a similar regularisation technique to make the variance-covariance matrix of internal variability more generalisable. Hence, our (ridge) regularisation step has a similar general purpose as in optimal fingerprinting (generalisability), however, is applied to a different aspect of the data (the signal in our case, the noise in optimal fingerprinting). Signal and noise are not explicitly separated in our method to estimate the $\beta$ fingerprint, but noise due to internal variability is accounted for implicitly by the regularisation step. Our method is also explained in detail in Sippel et al. (2020).

In addition to the similarities and differences w.r.t. optimal fingerprinting described above, we provide more context below, to situate our method in the landscape of well-known as well as less well-known, recently developed D&A methods.

The purpose of D&A - isolating the forced climate change signal in observations and comparing this to the forced response in climate models - requires separation of the signal and the noise. Several ways to do this have been developed, which, one could argue, all derive from Hasselman (1979). From here, optimal fingerprinting (e.g. Allen & Stott, 2003; Ribes et al., 2013a, 2013b) and detection methods based on pattern similarity (e.g. Santer et al., 2013, 2018; Marvel & Bonfils, 2013, 2020; Bonfils et al., 2021), sometimes called "non-optimal detection", have evolved. Our ridge regression based detection method is most closely related to pattern similarity methods as used in e.g. Santer et al. (2013). In Santer et al.'s work, the detection metric is defined as the projection of the variable of interest onto the leading empirical orthogonal function (EOF), derived from aggregated forced model simulations (i.e., an estimate of the signal pattern). In this method, the EOF-based signal pattern is referred to as the fingerprint, and the projection of observations and piControl simulations onto this pattern yields a one-dimensional detection metric, where detection is assessed as the deviation of the observations (e.g. trends of L years) from the distribution of unforced control simulations.

The ridge regression method described above and in our paper, builds on these pattern similarity detection methods in a straightforward way by adding a step in between signal pattern determination and projection. In our method, we project observations not onto the signal pattern directly, but onto a regression coefficient pattern that "optimally" (linearly, optimised by regularisation) maps Rx1d or PRCPTOT patterns onto a forced response metric based on the signal pattern. Hence, the ridge regression method can be seen as one step towards "optimising" the signal pattern (fingerprint) by increasing the signal-to-noise ratio (SNR).

We believe that the advantages of our method lie in (1) its relative simplicity and close links to the pattern similarity based D&A methods, while going beyond comparisons to the signal pattern (e.g. Marvel & Bonfils., 2013) or spatial aggregation techniques for Rx1d or PRCPTOT (Fischer et al., 2014; Donat et al., 2016), (2) the interpretable and relatively intuitive fingerprint map $\beta$ that reflects regions exhibiting high SNR climate change signals, (3) the actual estimate of the observed forced response time series resulting from application of this map to observations, so trends in this metric can be analysed, and (4) the possibility

to straightforwardly introduce additional constraints to, for instance, guard against specific climate uncertainties, such as uncertainty of whether climate models represent the correct magnitude of decadal-scale internal variability (Sippel et al., 2021). This method fits in recent developments in D&A that move towards mapping multidimensional data onto a one-dimensional detection space. Studies based on neural networks and deep learning for detection and attribution, e.g. Barnes et al. (2019, 2020), Labe & Barnes (2021), Madakumbura et al. (2021), Ham et al. (2022, preprint), employ non-linear methods - as opposed to our linear ridge regression method - but use a very similar framework with similar goals. We do not argue that ridge regression is fundamentally better than any of these older or newer methods, but we are convinced that the intuitive, physical outputs combined with high SNR can be valuable for trend detection and attribution.

We hope that these considerations address the reviewer's concerns. Based on this, we suggest to address issues raised in a revised paper by extending the Methods section with one paragraph to address in more detail the workings of our method, and relate it to existing D&A methods such as optimal fingerprinting and pattern similarity methods as explained above (in shortened and streamlined form). In addition, we will also make sure to eliminate redundant use of terminology commonly used in an optimal fingerprinting context, to avoid confusion about terms that have different meanings in our context. Nonetheless, we would prefer to continue to use the term "fingerprint" since this has been used in multiple contexts (optimal and pattern similarity fingerprinting) to describe the signature of a forcing in the climate, which is also how we use it (see the comparison with pattern similarity methods above). However, in order to prevent confusion we will make sure to add the side note that our method to create fingerprints differs from the optimal or pattern similarity methods.

Also, I think it is necessary for the authors to discuss whether the proposed methods, which basically use linear statistical models that therefore implicitly assume Gaussian, or near Gaussian errors, are suitable for the data to which they are applied. Indicators of extreme precipitation, such as Rx1day at individual grid boxes, are certainly not Gaussian.

Thanks. We fully agree that local distributions of Rx1d are not Gaussian and that methods assuming Gaussianness can not be applied to such distributions. However, we only use individual grid box time series of Rx1d as predictors in our linear regression model. Predictors in linear regression models need not be Gaussian as long as no parametric confidence intervals of regression coefficients are derived (which we don't do).

Below, we added Q-Q plots of the forced response estimates obtained by applying our detection fingerprint to piControl data (450 years per model). As can be seen, these unforced estimates are normally distributed, also for the Rx1d case. This assures the remnant natural variability in the forced response estimates is normally distributed.

Q-Q plots piControl PRCPTOT forced response estimates

[Figure]

Figure 1: Q-Q plots of PRCPTOT piControl forced response estimates obtained by applying the fingerprint to piControl simulations of 10 CMIP6 models, 450 years per model. The Q-Q plots are separated by model since different models may have different spread. When points lie on the diagonal slope-1 linear line, the distribution is normal.

Q-Q plots piControl Rx1d forced response estimates

[Figure]

Figure 2: As figure 1 but for Rx1d.

For the estimation of time of emergence, we implicitly assume the residuals of the linear fit of forced response estimates to GMST are normally distributed. In this application the local temporal distribution of Rx1d values is no longer of influence. We do not show the ToE residuals to be normally distributed, but we can add this to the supplementary information if so required.

A final general comment is that the relatively heavy of use of acronyms in this paper is not very reader friendly.

Point taken. We'll reduce acronym use where possible.

**References**

Allen, M.R., Stott, P.A. (2003). Estimating signal amplitudes in optimal fingerprinting, part I: theory. *Climate Dynamics* 21, 477–491 . https://doi.org/10.1007/s00382-003-0313-9

Ribes A., Planton, S., Terray, L. (2013). Application of regularised optimal fingerprinting to attribution. Part I : method, properties, and idealised analysis, *Climate Dynamics*, 41(11-12), 2817-2836, 10.1007/s00382-013-1735-7.

Ribes A., Terray, L. (2013). Application of regularised optimal fingerprinting to attribution. Part II : application to global near-surface temperature based on CMIP5 simulations, *Climate Dynamics*, 41(11-12), 2837-2853, 10.1007/s00382-013-1736-6.

Hasselmann, K. (1979). On the signal-to-noise problem in atmospheric response studies. In D. B. Shaw (Ed.), *Meteorology over the tropical oceans* (pp. 251-259). Bracknell: Royal Meteorological Society.

Santer, B. D., Painter, J. F., Mears, C. A., Doutriaux, C., Caldwell, P., Arblaster, J. M., ... & Zou, C. Z. (2013). Identifying human influences on atmospheric temperature. *Proceedings of the National Academy of Sciences*, *110*(1), 26-33.

Santer, B. D., Po-Chedley, S., Zelinka, M. D., Cvijanovic, I., Bonfils, C., Durack, P. J., ... & Zou, C. Z. (2018). Human influence on the seasonal cycle of tropospheric temperature. *Science*, *361*(6399), eaas8806.

Marvel, K., & Bonfils, C. (2013). Identifying external influences on global precipitation. *Proceedings of the National Academy of Sciences*, *110*(48), 19301-19306.

Marvel, K., Biasutti, M., & Bonfils, C. (2020). Fingerprints of external forcings on Sahel rainfall: aerosols, greenhouse gases, and model-observation discrepancies. *Environmental Research Letters*, *15*(8), 084023.

Bonfils, C. J., Santer, B. D., Fyfe, J. C., Marvel, K., Phillips, T. J., & Zimmerman, S. R. (2020). Human influence on joint changes in temperature, rainfall and continental aridity. *Nature Climate Change*, *10*(8), 726-731.

Sippel, S., Meinshausen, N., Fischer, E. M., Székely, E., & Knutti, R. (2020). Climate change now detectable from any single day of weather at global scale. *Nature climate change*, *10*(1), 35-41.

Fischer, E. M., Beyerle, U., & Knutti, R. (2013). Robust spatially aggregated projections of climate extremes. *Nature Climate Change*, *3*(12), 1033-1038.

Donat, M. G., Lowry, A. L., Alexander, L. V., O'Gorman, P. A., & Maher, N. (2016). More extreme precipitation in the world's dry and wet regions. *Nature Climate Change*, *6*(5), 508-513.

Sippel, S., Meinshausen, N., Székely, E., Fischer, E., Pendergrass, A. G., Lehner, F., & Knutti, R. (2021). Robust detection of forced warming in the presence of potentially large climate variability. *Science advances*, *7*(43), eabh4429.

Barnes, E. A., Hurrell, J. W., Ebert-Uphoff, I., Anderson, C., & Anderson, D. (2019). Viewing forced climate patterns through an AI lens. *Geophysical Research Letters*, 46(22), 13389– 13398. https://doi.org/10.1029/2019GL084944

Barnes, E. A., Toms, B., Hurrell, J. W., Ebert-Uphoff, I., Anderson, C., & Anderson, D. (2020). Indicator patterns of forced change learned by an artificial neural network. *Journal of Advances in Modeling Earth Systems*, *12*(9), e2020MS002195.

Labe, Z. M., & Barnes, E. A. (2021). Detecting climate signals using explainable AI with single-forcing large ensembles. *Journal of Advances in Modeling Earth Systems*, 13, e2021MS002464. https://doi.org/10.1029/2021MS002464

Madakumbura, G.D., Thackeray, C.W., Norris, J., Goldenson, N., Hall, A. (2021). Anthropogenic influence on extreme precipitation over global land areas seen in multiple observational datasets. *Nat Commun* 12, 3944. https://doi.org/10.1038/s41467-021-24262-x

Ham, Y.-G. Kim, J.-H., Min, S.-K. Kim, D., Li, T., Timmermann, A., Stuecker, M. Anthropogenic fingerprints in daily precipitation revealed by deep learning, 03 August 2022, PREPRINT (Version 1) available at Research Square [https://doi.org/10.21203/rs.3.rs-1860132/v1]

---

## Author Comment (AC2)

Comments of Referee #2
Response to Referee #2

Overall comments:

This study conducts a signal detection analysis for global changes in mean and extreme precipitation using three observational datasets and CMIP6 multi-model outputs. The authors apply a ridge regression (RR) method to construct fingerprints, which helps increase a signal-to-noise ratio of precipitation change patterns. Results show a robust detection of anthropogenic signals in all observations for both mean and extreme precipitation even when removing global mean trends, further supporting the human-induced intensification of global hydrological cycle. I find this paper very well written with sufficient details provided about methods as well as various sensitivity tests and therefore suggest publication after addressing some minor issues.

Thanks very much for your kind comments and positive judgment of our manuscript.

Major comments:

1. Although method details are provided, it would be useful to explain more clearly what are benefits of the attribution approaches employed, including ridge regression, EOF-based metric for target variable, and GMST-based signal estimation. All of these procedures seem to contribute to increase signal-to-noise ratio but how they do and what step is more important. The authors provide some associated results from sensitivity tests but an overall explanation of their method possibly with a schematic would be helpful for readers to understand the contribution of each step to the final signal detection.

We see that the sequence of steps and their relative function with respect to one another can lead to confusion. We like the idea of adding a schematic of the methodology, and we will add this to the supplementary info of a revised paper. We show preliminary drafts of such flowcharts below.

**Flowchart part I: Schematic visualisation of determination of ridge regression targets**

[Figure]

**PRCPTOT or Rx1d anomalies full coverage**

Different models

Ensemble member

**Single model mean anomalies**

**Multi-model mean anomalies**

Principal component analysis

**First empirical orthogonal function (EOF) and principal component (PC) of multi-model mean anomalies**

Projection of single model means onto multi-model mean EOF

**Model specific forced response targets** $Y$

**Flowchart part II: Schematic visualisation of ridge regression procedure and determination of observed forced response estimates**

**Targets: PRCPTOT or Rx1d model projections of multi-model mean EOF** $Y$

**Predictors: PRCPTOT or Rx1d anomalies masked to observational coverage** $X$

Different models

Ensemble member

$\sim$

Ridge regression

**Detection fingerprint** $\beta$

Apply to observations

**Observations** $\hat{X}$

$\hat{X} \beta$

Different observational datasets

**Observed forced response estimates** $\hat{Y}$

In addition, we add a figure to the supplementary information which allows comparison of the signal-to-noise ratios (SNR) of the procedure performed with

1. EOF based targets (our chosen default) and "optimal" regularisation ($\lambda\_sel$)
2. Global mean based targets and "optimal" regularisation
3. EOF based targets and minimal regularisation ($\lambda\_0$).

Comparing 1 and 2 gives an impression of the SNR-effect of using EOF based targets, whereas comparing 1 and 3 shows the SNR-effect of ridge regression (relative to unregularised ordinary least squares).

[Figure]

Figure 1: SNR of PRCPTOT (left) and Rx1d (right) forced response estimates from all observational datasets, regressed onto smoothed global mean surface temperature (GMST) (as in manuscript), for cases 1, 2, and 3 as above.

Comparing cases 1 and 2: As can be seen in figure 1, the SNR does not necessarily increase by using the EOF based target instead of the global mean target; for PRCPTOT, the EOF based target exhibits lower SNR, whereas for Rx1d, it does not make any difference whether we use the global mean based target or the EOF based target. The choice of using the EOF based metric for PRCPTOT thus requires some explanation. The global mean based target leads to higher SNR because the trend in global mean precipitation is stronger than the trend in the first EOF of mean precipitation, and models are more in agreement on global mean precipitation change. However, since forced changes in mean precipitation behave according to a pattern of wetting and drying regions (e.g. Held & Soden (2006)), the global mean trend in precipitation is not a very refined measure of forced precipitation changes. The first EOF captures the forced pattern of change, and its corresponding principal component time series captures the strength of that pattern. The first principal component is thus a reflection of the forced pattern strength (e.g. Marvel & Bonfils, 2013), meaning the forced response in all regions is somewhat reflected in this timeseries, and not averaged out as in the global mean. In addition, individual models' deviations from the multi-model

pattern due to uncertainties in e.g. the forced response in circulation, are reflected in the projections of the EOF on the model ensemble means which serve as our model-specific forced response targets. We argue that including the uncertainties in the forced response, reflected by uncertainties in the first principal component, has preference and may prevent overconfident detection of a signal. We argue this is a more balanced reflection of the forced response.

Since the EOF-based target metric has a weaker trend and more variability for PRCPTOT, the ridge model and the forced response estimates are "pushed" in a more conservative direction. We argue that this is the better approach, given that the goal is not to construct a ridge model that generates the strongest forced response estimate, but one that is most likely to predict the true forced response given the observations that are available. We therefore use the more conservative estimates, which implicitly include pattern information and uncertainties, by default. We point out, however, that the main conclusions, which are detection of a forced response but disagreement among observational datasets on the observed forced response relative to the simulated forced response, are insensitive to the choice of target metric.

Comparing cases 1 and 3: This comparison indicates the benefit of using regularised regression. $\lambda\_0$ is not equivalent to ordinary least squares, in that $\lambda$ is not set to 0, but it is the smallest $\lambda$ used in the training procedure, and in all cases at least two orders of magnitude smaller than $\lambda\_sel$. A smaller $\lambda$ increases the variability in the forced response estimate, but, likely, also the trend. Therefore, when it comes to SNR, the effect of $\lambda$ is a trade-off between the increased variability and the increased trend. For Rx1d, we see that a smaller $\lambda$ deteriorates the detectability → overfitting leads to large variability increase without reducing a low trend bias. In PRCPTOT, the effect is messier. For HadEX3, the SNR clearly decreases for smaller $\lambda$, but for GHCNDEX and GPCC this is not the case. Analysis shows that the strong uptick at the end of the GHCNDEX record (referred to in L283 of the manuscript) is somewhat dampened by larger $\lambda$s. When $\lambda$ is minimised, the GHCNDEX forced response estimate shows this strong increase in the last few years of the record, which amplifies the overall trend, and therefore high SNRs are seen. For this $\lambda$, however, physical consistency of the fingerprints is strongly impaired, as can be seen below, comparing $\lambda\_sel$ and $\lambda\_0$.

[Figure]

PRCPTOT RR coefficients [−]

< −8e−04    −4e−04    0    4e−04    > 8e−04

Figure 3a: GHCNDEX detection fingerprint for $\lambda\_sel$

[Figure]

PRCPTOT RR coefficients [−]

< −8e−04    −4e−04    0    4e−04    > 8e−04

Figure 3b: GHCNDEX detection fingerprint for $\lambda\_0$

For GPCC, the low coverage leads to generally very high variability in the forced response estimate, as also witnessed by the low SNRs. A smaller $\lambda$ leads to a slightly larger increase in trend relative to the increase in variability, however, the fingerprints no longer reflect any physical consistency, as shown below. Polson et al. (2013) also found it is difficult to detect forced responses in GPCC.

[Figure]

Figure 4a: GPCC detection fingerprint for $\lambda\_sel$

Figure 4b: GPCC detection fingerprint for $\lambda\_0$

The above shows that it is important to assess the complete result of fingerprints, forced response estimates, and SNRs to judge the quality of the detection model and the detected response. PRCPTOT is generally a more difficult variable to detect forced trends in, due to the spatial pattern of change and high internal and model variability in the representation of this pattern. This was also found by e.g. Fischer & Knutti (2014). For the most recent, higher-resolution and higher-coverage HadEX3 dataset, however, ridge regression also has clear benefits for the detection of forced trends in PRCPTOT, besides the fingerprint interpretability advantages which we see in all three observational datasets.

2. An important motivation of considering different periods and datasets is opposing conclusions by previous studies about model overestimation or underestimation of the observed trends. I am wondering if the authors can go further and compare their results with some previous studies. For instance, if studies based on the latter half of 20th century trends find model underestimation, the authors can assess their model trends for the same/similar periods. Another point here is that the present study uses absolute units of precipitation while most of previous studies considered relative changes or aggregated values. It would be good to discuss possible influences of this difference.

Thanks for these very valid comments. We identify two main comments here - one being the comparison with previous studies and the other being the comparison between absolute versus relative units of precipitation. We address these two issues separately below, in reverse order.

*Comparison of different precipitation metrics*

To address this, we intend to add a section to the supplement with a concise description of the comparison presented below.

Some studies define precipitation change as a percentage change relative to climatological precipitation levels per degree of global temperature change. One can determine relative precipitation changes at the gridpoint level (normalised w.r.t. climatological gridpoint-mean precipitation) or at the global level (most common - underlying numbers such as ~2%/K and ~7%/K for PRCPTOT and Rx1d). Determining relative precipitation changes at the gridpoint level ensures that e.g. the tropics - a region with high absolute precipitation changes due to high climatological precipitation (Clausius-Clapeyron) - do not dominate the overall response. However, it could also lead to inflation of trends at grid points with very low climatological precipitation levels (e.g. desert areas into which precipitating bands shift, where local relative precipitation metrics approach infinity due to dividing by close-to-zero climatological levels), which is why we do not use gridpoint-level relative precipitation change.

Whereas we thus use absolute units in our predictors, our forced response metric (the model projections onto the first multi-model mean EOF) does not have meaningful physical units, but reflects a time series that includes pattern information, as mentioned above (it is a linear transformation of the raw data in original units). Implicitly, this already partially accounts for the regional differences in the expected absolute trends, since the pattern has higher loading in regions where precipitation is climatologically high. Note also that our forced response estimates do not have meaningful physical units in terms of mm/s, and reflect the strength of the forced response pattern, rather than the absolute change in precipitation in mm. Nonetheless, differences in overall precipitation level between different models and observations, which can affect the found strength of the forced pattern, are not accounted for.

Hence, to allow comparison with studies that use *global* relative precipitation change in % (which do not suffer from the approach toward division-by-zero that can occur at some gridpoints when *local* relative change is used), we have normalised our forced response estimate trends and model target trends with respect to their corresponding global mean precipitation levels (average over the gridpoints in the observational masks). We assess the model forced response *targets* (EOF-based) and the observational forced response *estimates*, since this allows assessing whether the answer to the question "do models over- or underestimate observed forced change?" depends on the unit of precipitation (absolute vs. normalised). Note that we normalise our forced response estimate trends, which are unitless. The resulting trend unit is thus $mm^{-1}$.

Figure 5 shows the results. Note that these plots represent three points (start years 1951, 1971, and 1991, from left to right) in Figure 2 in the manuscript. The different start years, as in the manuscript, allow for assessment of changing relative trends depending on trend period. Comparing the left and right half of each plot reveals the difference between the original trends as in the manuscript (left) and the normalised ones (right).

For PRCPTOT, we see that normalising trends w.r.t. climatological mean precipitation shifts the modelled forced trends down relative to observations, consistent with the models exhibiting slightly higher climatological PRCPTOT levels - a known persistent systematic bias (e.g. Stephens et al., 2010). Despite slight decreases in model forced trends, it remains the case that the relative magnitude of model forced trends and observed forced trend estimates depends on the period and observational dataset.

[Figure]

Figure 5a: Comparison of original PRCPTOT trends (as in manuscript) and trends normalised by the model's/observation's corresponding climatological PRCPTOT level; the 1951-2014 mean, averaged over the observational masks. Trends of single-model targets (points and corresponding boxplot indicating the interquartile range), and observed forced response estimates (X-marks). Non-physical units, black dashed line indicates 0.

[Figure]

Figure 5b: As 5a but for Rx1d

For Rx1d (Figure 5b), on the contrary, normalising trends w.r.t. climatological mean Rx1d increases forced model trends relative to observed forced trend estimates, suggesting climatological mean levels of Rx1d are lower in models than in observations, which is also a known model bias (e.g. Sillmann et al., 2013, Bador et al., 2020). Nonetheless, again, main conclusions on the relative model vs. observational trend magnitudes do not change. These opposing findings regarding PRCPTOT and Rx1d, align well with the findings of Fischer & Knutti (2016), who suggest PRCPTOT changes are overestimated by models, whereas Rx1d changes are underestimated.

Some studies assess precipitation change as a function of global mean temperature change, e.g. in %/K. Given the relationship between temperature and specific humidity/saturation vapour pressure (Clausius-Clapeyron), this can in fact make a large difference, since different models, as well as observations, warm at different rates (different climate sensitivity). Although our forced response metrics, as said above, represent strength of forcing, we can still normalise the strength of forcing w.r.t. global mean warming to account for differences in climate sensitivity.

Therefore, we further normalise the relative trends shown above, by dividing by the temperature change over the trend period. This results in trends that are independent of the model's and observations' differences in climatological precipitation levels and warming rate (climate sensitivity).

Model targets are normalised w.r.t. their specific model's mean global mean surface air temperature (GSAT) change over the corresponding trend period, and observational forced responses are normalised w.r.t. the GMST change from the Cowtan & Way (2014) temperature dataset. We determine GMST change by simply computing the difference between the 2020 value and the values in 1951, 1971, and 1991 of the 21-year LOWESS-smoother GMST.

The comparison between original trends, as in the manuscript, and relative GMST-normalised trends (in $mm^{-1}K^{-1}$) is shown in the figures below for PRCPTOT (6a) and Rx1d (6b). Comparing the left and right column in each panel shows that normalising the forced relative trends from Figure 5 w.r.t. their corresponding temperature change reduces model spread, which is to be expected. For PRCPTOT (Figure 6a), GMST-normalisation further reduces model trend magnitude relative to observed forced trend estimates, since model warming rate in CMIP6 is higher than in observations. Therefore, for Rx1d (Figure 6b), GMST-normalisation reduces model trends as well, and offsets some of the effect of normalising w.r.t climatological Rx1d levels seen in figure 5b.

However, more importantly, figure 5 and 6 show that, compared to the original trends, the relative magnitude of model and observational trends changes somewhat in response to normalising w.r.t climatology and warming rate, but the main picture does not change - relative trend magnitudes still differ between periods and observational datasets. The main conclusion of our study – forced trends are detected, but observations lie on different ends of the model-projected spectrum – holds also for normalised trends.

[Figure]

Figure 6a: Comparison of original PRCPTOT trends (as in manuscript) and trends normalised by the model's/observation's corresponding climatological PRCPTOT level; the 1951-2014 mean, averaged over the observational masks. Trends of single-model targets (points and corresponding boxplot indicating the interquartile range), and observed forced response estimates (X-marks). Non-physical units, black dashed line indicates 0.

[Figure]

Figure 6b: As 6a but for Rx1d

The table below contains an overview of whether the model trend interquartile range lies higher than (+) lower than (-), or contains (0) each observational time series. As can be seen, the full range from under- to overestimation of observed trends by models is covered across trend periods and observational datasets, both for original as well as normalised trends.

| Obs dataset | Trend period | PRCPTOT | | Rx1d | |
|---|---|---|---|---|---|
| | | original [-] | norm [-/mm/K] | original [-] | norm [-/mm/K] |
| HadEX3 | 1951-2020 | 0 | - | 0 | 0 |
| | 1971-2020 | 0 | - | 0 | 0 |
| | 1991-2020 | + | 0 | 0 | + |
| GHCNDEX | 1951-2020 | - | - | - | - |
| | 1971-2020 | - | - | 0 | - |
| | 1991-2020 | - | - | 0 | + |
| GPCC | 1951-2020 | 0 | 0 | | |
| | 1971-2020 | + | + | | |
| | 1991-2020 | + | + | | |

In addition, to complete the comparison, we assessed trends in %/K (i.e. physical units), which are obtained by using the forced response estimates based on the ridge model with the *global mean* target (supplementary information section S2.2). The main conclusion still does not change qualitatively - relative model and observational trends remain dependent on observational dataset and trend period. Normalising even suggests larger differences across different observational datasets.

This check also shows that global mean based ridge regression also reproduces numbers in the range of the well-known 2-3%/K change in global mean PRCPTOT. For Rx1d, the ~5%/K change we find is lower than the ~7%/K change prescribed by Clausius-Clapeyron, which has been found for CMIP models of different generations before (e.g. Allan & Soden, 2008, Kotz et al. (preprint)). Note that we are restricted to normalising with respect to a climatological precipitation value that is based on the mean over the grid cells with observational coverage, in order to "treat" model and observational data the same. Therefore, the percentages may be off, since the global mean differs from the mean we use. (Note - these numbers only apply to global mean changes, not to local gridpoint changes.)

[Figure]

Figure 7a: Comparison of original PRCPTOT trends (as in manuscript) and trends normalised by the model's/observation's corresponding baseline (1951-2014) PRCPTOT level and global mean surface temperature (GMST) change for three different periods (1951-2020, 1971-2020, 1991-2020). Target trends (points/boxplot) and forced response estimates are based on the global mean in this case, leading to physical units.

[Figure]

Figure 7b: As 7a but for Rx1d.

*Comparison to previous studies*

We suggest addressing this by adding a few relevant references to previous studies in the discussion of the results, as well as by adding a table to the supplementary information. This table contains an overview of a set of recent papers that have attempted detection of either PCRPTOT or extreme indices (Rx1d or e.g. Rx5d) and that have made assertions on model vs. observational trends. This provides an overview of the (dis)agreement regarding over- or underestimation of observed forced changes in precipitation by models. See a preliminary version of the table below.

From this table, several interesting comparisons result. First of all, scaling factors obtained through optimal fingerprinting for PRCPTOT often lead to the result that models underestimate observed change. Assessing the spatial distribution of trends, however, shows that models in fact produce positive PRCPTOT trends over a larger land fraction than observations do. In our study, models also underestimate PRCPTOT trends in GHCNDEX, but agree better with, or overestimate, HadEX3. HadEX3 has considerably higher coverage and resolution than GHCNDEX, and also than HadEX2. Also GPCC estimates much lower global forced response trends than model projections, whereas Knutson & Zeng (2018) find higher local trends in GPCC compared to CMIP5 models. Model underestimation of local internal variability in mean precipitation may partly cause this. This also aligns well with the findings of Fischer & Knutti (2014), since they assess trends in units of local standard deviation. The higher PRCPTOT trends in models may be an artefact of underestimation of local PRCPTOT variability (standard deviation) in models.

For Rx1d, optimal fingerprinting studies often use a probability index (PI), meaning effectively that trends in percentiles are assessed (an increasing prevalence of Rx1d values that lie further to the right on the local GEV-distribution of Rx1d values). An interesting finding is that this approach leads to the conclusion that models overestimate Rx1d changes based on scaling factors, whereas normalising changes by warming rate and computing trends, leads to the conclusion that models underestimate trends in Rx1d with warming (Paik et al, 2020). We showed above that our method does not lead to fundamentally different results depending on the metric used (non-normalised changes or relative changes as a function of warming). Primarily, we can conclude that the observational dataset seems to have a large influence on the results. Where e.g. GPCC did not allow detection in any of the assessed cases in Polson et al. (2013), GHCNDEX often seems to suggest models underestimate observations. HadEX3, with highest coverage and resolution, lies in between these two extremes in our study. Overall, these comparisons suggest that observational uncertainty is still large and may be highly relevant as to whether models over- or underestimate precipitation trends. This is consistent with Bador et al. (2020), who find that observational uncertainty can be partly as large as uncertainty across climate models.

| Paper | Model archive | Obs dataset | Spatial region | Variable | Method | Trend periods | Models w.r.t. observations? | Remarks |
|---|---|---|---|---|---|---|---|---|
| **PRCPTOT** | | | | | | | | |
| Noake et al. (2012) | CMIP3 | GHCN, CRU, VASClimO | Global land, separated into 5deg latitude bands. Scaling factors determined for spatiotemporal aggregate, not per latitude band | Seasonal PRCPTOT percentage change per latitude band | Optimal fingerprinting | 1952-1990, 1960-1999, 1951-1990, 1975-1999 periods | - | "-" applies to scaling factor best estimate for seasons and observational datasets in which significant change is detected (confidence interval does not include 0), and holds for all trend periods |
| Wu et al. (2013) | CMIP5 | GHCN | Northern hemisphere land | PRCPTOT percentage change | Optimal fingerprinting | 1952-2011 | - | |
| Polson et al. (2013) | CMIP5 | GHCN, CRU, VASClimO, GPCC | Global land, separated into 5deg latitude bands. Scaling factors determined for spatiotemporal aggregate, not per latitude band | Seasonal PRCPTOT percentage change per latitude band | Optimal fingerprinting | 1951-2005 (2000 for VASClimO) | - | Applied Noake's method to CMIP5, "-" applies to scaling factor best estimate for seasons and observational datasets in which significant change is detected (confidence interval does not include 0), GPCC never shows a detectable climate signal. |
| Fischer & Knutti (2014) | CMIP5 + CESM initial condition ensemble | HadEX2, GHCNDEX | Global | Spatial distribution of gridpoint trends in PRCPTOT, expressed in terms of local sigma (based on 1986-2005 interannual variability) | Spatial probability distribution comparison | 1960-2010 | + | Models estimate more regions with positive trends in PRCPTOT, but not enough negative trends --> too much wetting |
| Knutson & Zeng (2018) | CMIP5 | GPCC | Global, per gridpoint | Linear trend in PRCPTOT | Linear trend fitting to grid point timeseries | 1901-2010, 1951-2010, 1981-2010 | - | Models cannot produce the magnitude of positive nor negative trends in obs. Discrepancy gets stronger in later trend periods |
| **Rx1d** | | | | | | | | |
| Min et al. (2011) | CMIP3 | HadEX | NH land, separated into (overlapping) regions: mid-latitudes, tropics | Rx1d and Rx5d Probability index: 0-1 quantile per gridpoint | Optimal fingerprinting | 1951-1999 | - | "-" applies to scaling factor best estimate for regions where there is detection |
| Zhang et al. (2013) | CMIP5 | HadEX2 + russian station data | NH land, separated into (overlapping) regions: Western Eurasia, Eastern Eurasia, North America, mid-latitudes, tropics | Rx1d and Rx5d Probability index: 0-1 quantile per value, based on fit GEV per gridpoint/station and then interpolated | Optimal fingerprinting | 1951-2005 | 0/+ | Scaling factor estimates include 1, but best estimates are still below 1 |
| Fischer & Knutti (2014) | CMIP5 + CESM initial condition ensemble | HadEX2, GHCNDEX | Global | Rx5d, expressed in terms of local sigma (based on 1986-2005 interannual variability) | Spatial probability distribution comparison | 1960-2010 | - | Models don't show a large enough land fraction exhibition positive trends, and do not reproduce the magnitude of the largest trends seen in observations |
| Fischer & Knutti (2016) | CMIP5 and EURO-CORDEX | E-OBS/Ensembles | Europe | Changing occurrence of historical >90 percentile values of daily precipitation | Probability distribution comparison | 1951-1980 and 1981-2013 distributions | - | Models show smaller increase in intensity of >90th percentile daily precipitation amounts |
| Borodina et al. (2017) | CMIP5 + CESM initial condition ensemble | GHCNDEX, HadEX2 | Global land, selected wet regions only (wettest 40%, agreed across models) | Rx1d percentage change per gridpoint as a function of GMST [%/K], averages over wet regions, as well as land area fraction experiencing positive Rx1d trends | Trend comparison | 1951-2005 | - | Models show smaller trends than both observational datasets, but HadEX2 shows smaller trends than GHCNDEX |
| Paik et al. (2020) | CMIP5 | HadEX2 | Global land, separated into (overlapping) regions: Western Eurasia, Eastern Eurasia, North America, mid-latitudes, tropics, wet and dry regions. | Rx1d and Rx5d Probability index: 0-1 quantile per value, based on fit GEV per gridcell/station and then interpolated. Scaling factors | Optimal fingerprinting | 1950-2020 | 0/+ | "0" applies to EU and dry regions, where models and observations agree. For all other regions with detection, models overestimate the change ("+") |
| Paik et al. (2020) | CMIP5 | HadEX2 | Global land, separated into (overlapping) regions: Western Eurasia, Eastern Eurasia, North America, mid-latitudes, tropics, wet and dry regions. | Rx1d and Rx5d Probability index: 0-1 quantile per value, based on fit GEV per gridcell/station and then interpolated. Spatially averaged trends, normalised by GMST | Trends in %/K | 1950-2020 | - | Note: same study as above. In all regions where forced change is detected, models underestimate observations when trends are assessed. In these same regions, scaling factors suggest that models overestimate change. |
| Sun et al. (2022) | CMIP6 and CanESM2 LE | HadEX2 stations + russian and chinese station data | Global, continental, regional | Rx1d and Rx5d, Non stationary spatiotemporal varying GEV-based optimal fingerprinting, no normalisation: absolute units of precipitation (log) | Non-optimal variant of optimal fingerprinting: scaling factor determination but no internal variability covariance corrections | 1950-2014 | + | "+" applies to all continents/regions, and also global level, but Northwestern Europe (Scandinavia/UK) where scaling factors are around 1 ("0") |

Table 1: Previous D&A studies on PRCPTOT and Rx1d, including their main findings on whether modelled forced changes are smaller (-), similar (0) or larger (+) than observed forced changes

3. The lower detectability in GHCNDEX observations are suggested to be due to the poorer spatial coverage. Regarding this issue, I would suggest using Rx5d. As I understand, Rx5d has larger spatial coverage than Rx1d and comparison with Rx1d-based results may provide a way to support the authors' interpretation. Another way would be to compare detection results from using a selected model run but with different spatial coverages applied.

We assessed the effect of coverage by masking HadEX3 as GHCNDEX and running the ridge regression and detection procedure. This resulted in higher consistency between HadEX3 and GHCNDEX, but did not reconcile the differences fully. Therefore, coverage only explains part of the differences. We will make this more clear in the text (L318-320).

Also, our primary motivation for using PRCPTOT and Rx1d is to assess mean and extreme precipitation separately; Rx5d would not accomplish this goal as well as Rx1d because it is less extreme and thus more similar to PRCPTOT (Pendergrass & Knutti, 2018). Furthermore, to our knowledge, Rx5d does not have higher coverage in GHCNDEX than does Rx1d, see below: all the white cells have no coverage. The difference between 1951-2020 coverage of GHCNDEX for Rx5d versus Rx1d is shown in figure 7c - the dark red cells have coverage for Rx5d but not in Rx1d, yellow cells have coverage for both. Given that the coverage increase is minor and only in areas where there is reasonable coverage already, we anticipate that this would not make much of a difference.

[Figure]

copyright www.climdex.org, 2022-09-29
10.1175/BAMS-D-12-00109.1

copyright www.climdex.org, 2022-09-29
10.1175/BAMS-D-12-00109.1

Figure 8a: Rx1d trend from climdex.org, white cells have no coverage

Figure 8b: Rx5d trend from climdex.org, white cells have no coverage

[Figure]

Figure 8c: Coverage differences GHCNDEX Rx5d and Rx1d: red cells are added in Rx5d w.r.t Rx1d.

Minor comments:

L8: Indicating analysis period or trend period with signal detection would be useful here.

In a revised manuscript we will changes this to "[...] to assess the degree of forced change detectable in the real-world climate in the period 1951-2020."

L17-19, L58-64: Better comparisons can be made by applying the same periods as those used in previous studies. See my major comment above.

See reply to major comment above: both previous studies as well as we assess multiple trend periods. Disagreements across studies and observational datasets remain.

L20-21: Is this confirmed by repeating detection analysis using NH-extratropics only?

Yes, see supplementary information.

L34: "discrepancies with respect to observations". Its meaning is unclear.

We hope this will be resolved by changing the sentence to "[...] model representation of the water cycle has also been shown to disagree with observations."

L69-71: Need to explain what the previous studies have found additionally using these "data-science methods". Also, what's the novelty of this study compared with them? Is it detection based on spatial pattern information alone?

As the sentence reads: these studies have detected forced signals. Since the main purpose of the D&A field is to answer the question "can we detect and attribute effects of forcing in observations", this is the finding that matters. Our method fits in these recent data science developments in D&A that move towards mapping multidimensional data onto a one-dimensional detection space. Studies based on neural networks and deep learning for detection and attribution, employ non-linear methods - as opposed to our linear ridge regression method - but use a very similar framework with similar goals. We do not argue that ridge regression is fundamentally better than any of the older or newer methods, but we are convinced that the intuitive, physical outputs combined with high SNR can be valuable for trend detection and attribution. (See also response to referee #1.)

L108-109: "Trend biases due to this structural difference … negligible". But the cited reference considered south-east Australia only?

This is true, however, the study investigates the effects of data operations on time series. The temperature and precipitation time series of course differ per region, but the effects of operations on the long term trends is not expected to differ greatly from region to region. Nonetheless, we will add the reference to Dunn et al. (2020) (also referenced in L107), who also makes this statement more generally.

L201: How to define S when global means are removed?

The definition of S stays the same. To obtain the results for the detrended case, global means are removed from the predictor data in training of the ridge model. The ridge model is however still trained to predict the same forced response target. Therefore, the forced response estimates, based on the detrended observations, still contain the forced trend (if the method works). These forced response estimates are regressed onto GMST to obtain S in the same way as for the default case. We hope this clarifies things.

L212: "CMIP6 ssp245" should be "CMIP6 historical"?

Yes, thanks for noticing this.

L227: "virtually identical". adding spatial correlation would help with this.

That is a good point, we will add the correlation value in a revised manuscript. Pearson correlations between linear trends and EOFs over the full historical-ssp245 period for both PRCPTOT and Rx1d, on both the coarser GHCNDEX and the finer HadEX3 grid, are > 0.99.

L314-316: This suggests possible dependence of Rx1d FRE on temperature, resembling global warming slowdown due to PDO influence?

Potentially, although we do not have enough evidence to claim that the levelling off of the trends is not simply due to shorter trend length and internal variability. Attributing changes in trend slope to lange scale modes of variability is outside the scope of this study.

L331-332: "results … hold when the global mean is used as FR target". Then what are benefits of using EOF-based metric for target variable?

See above discussion of using the EOF-based target metric. We will add a sentence or two to the method section to further justify this choice of target.

L382-383: "accuracy of the CMIP6 climate models in simulating the processes …". It's unclear how the authors get this conclusion. Observation-model agreement in residual variability? More explanation would be useful.

Precisely. As we mention in the manuscript, removing the mean trend from the predictor data implies that only the relative pattern of precipitation can be used by the ridge regression model to predict the forced trend (note: the area-mean trend is removed from every grid point time series, meaning that spatial pattern information (how local precipitation differs from the area-mean) is retained in the predictors). The ridge regression detection model is trained to predict the (model) forced response from simulated spatial precipitation patterns, meaning it finds the *simulated* relationship between spatial precipitation patterns and forced precipitation change. Applying the detection model to observations, results in detection of a forced trend in observations, which implies that the relationship between spatial precipitation patterns and the forced response that the ridge model learnt from models, also holds in observations. This thus implies accuracy of the models in representing the spatial patterns related to the forced response. We hope changing the last sentence and splitting it into two as below solves the confusion:

" Taken together, the above shows, first, detection of forced change in mean and extreme precipitation beyond a global mean trend, second, the power of RR for signal extraction from high-dimensional noisy data, and third, the accuracy of the CMIP6 climate models in simulating the processes relevant to the spatial pattern of forced change in mean and extreme precipitation." → " Taken together, the above shows, first, detection of forced change in mean and extreme precipitation beyond a global mean trend, and second, the power of RR for signal extraction from high-dimensional noisy data. Also, the fact that the relationship between relative spatial precipitation patterns and the forced precipitation trend learnt from climate model simulations by the ridge model holds in observations, suggests accuracy of the CMIP6 climate models in simulating the processes relevant to the spatial pattern of forced change in mean and extreme precipitation."

L394-395: "(not shown)". This looks important and I suggest showing them in the supplement.

We will add these plots to the supplementary information.

L428: "value of RR-based fingerprint construction". What happens in detection or SNR without applying RR? See my major comment above.

See reply to major comment above. Overfitting leads to high variability in forced response estimates, and low SNRs.

**References**

Held, I. M. and Soden, B. J.: Robust Responses of the Hydrological Cycle to Global Warming, Journal of Climate, 19, 5686 – 5699, https://doi.org/10.1175/JCLI3990.1, 2006.

Marvel, K. and Bonfils, C.: Identifying external influences on global precipitation, Proceedings of the National Academy of Sciences, 110, 19 301–19 306, https://doi.org/10.1073/pnas.1314382110, 2013

Fischer, E. M. and Knutti, R.: Detection of spatially aggregated changes in temperature and precipitation extremes, Geophysical Research Letters, 41, 547–554, https://doi.org/10.1002/2013GL058499, 2014.

Stephens, G. L., L'Ecuyer, T., Forbes, R., Gettelmen, A., Golaz, J.-C., Bodas-Salcedo, A., Suzuki, K., Gabriel, P., and Haynes, J.: Dreary state of precipitation in global models, *J. Geophys. Res.*, 115, D24211, doi:10.1029/2010JD014532, 2010.

Sillmann, J., Kharin, V. V., Zhang, X., Zwiers, F. W., and Bronaugh, D.: Climate extremes indices in the CMIP5 multimodel ensemble: Part 1. Model evaluation in the present climate, *J. Geophys. Res. Atmos.*, 118, 1716– 1733, doi:10.1002/jgrd.50203, 2013.

Fischer, E. M. and Knutti, R.: Observed heavy precipitation increase confirms theory and early models, Nature Climate Change, 6, 986–991, https://doi.org/10.1038/nclimate3110, 2016.

Cowtan, K. and Way, R. G.: Coverage bias in the HadCRUT4 temperature series and its impact on recent temperature trends, Quarterly Journal of the Royal Meteorological Society, 140, 1935–1944, https://doi.org/10.1002/qj.2297, 2014.

Allan, R. P., & Soden, B. J.: Atmospheric warming and the amplification of precipitation extremes. *Science*, *321*(5895), 1481-1484, 2008.

Kotz, M., Wenz, L., Lange, S., and Levermann, A.: Changes in mean and extreme precipitation scale universally with global mean temperature across and within climate models, https://doi.org/10.31223/X5C631, 2022 (preprint).

Bador, M., Boé, J., Terray, L., Alexander, L. V., Baker, A., Bellucci, A., et al. Impact of higher spatial atmospheric resolution on precipitation extremes over land in global climate models. *Journal of Geophysical Research: Atmospheres*, 125, e2019JD032184. https://doi.org/10.1029/2019JD032184, 2020.

Pendergrass, A. G., & Knutti, R.: The uneven nature of daily precipitation and its change. *Geophysical Research Letters*, 45, 11,980– 11,988. https://doi.org/10.1029/2018GL080298, 2018.

Dunn, R. J. H., Alexander, L. V., Donat, M. G., Zhang, X., Bador, M., et al.: Development of an 520 Updated Global Land In Situ-Based Data Set of Temperature and Precipitation Extremes: HadEX3, Journal of Geophysical Research: Atmospheres, 125, https://doi.org/10.1029/2019JD032263, 2020.

**References in table**

Noake, K., Polson, D., Hegerl, G., and Zhang, X.: Changes in seasonal land precipitation during the latter twentieth-century, Geophysical Research Letters, 39, https://doi.org/10.1029/2011GL050405, 2012.

Wu, P., Christidis, N., and Stott, P.: Anthropogenic impact on Earth's hydrological cycle, Nature Climate Change, 3, 807–810, https://doi.org/10.1038/nclimate1932, 2013.

Polson, D., Hegerl, G. C., Zhang, X., and Osborn, T. J.: Causes of Robust Seasonal Land Precipitation Changes, Journal of Climate, 26, 6679 – 6697, https://doi.org/10.1175/JCLI-D-12-00474.1, 2013.

Knutson, T. R. and Zeng, F.: Model Assessment of Observed Precipitation Trends over Land Regions: Detectable Human Influences and Possible Low Bias in Model Trends, Journal of Climate, 31, 4617 – 4637, https://doi.org/10.1175/JCLI-D-17-0672.1, 2018

Min, S.-K., Zhang, X., Zwiers, F. W., and Hegerl, G. C.: Human contribution to more-intense precipitation extremes, Nature, 470, 378–381, https://doi.org/10.1038/nature09763, 2011.

Zhang, X., Wan, H., Zwiers, F. W., Hegerl, G. C., and Min, S.-K.: Attributing intensification of precipitation extremes to human influence, Geophysical Research Letters, 40, 5252–5257, https://doi.org/10.1002/grl.51010, 2013

Borodina, A., Fischer, E. M., and Knutti, R.: Models are likely to underestimate increase in heavy rainfall in the extratropical regions with high rainfall intensity, Geophysical Research Letters, 44, 7401–7409, https://doi.org/10.1002/2017GL074530, 2017.

Paik, S., Min, S.-K., Zhang, X., Donat, M. G., King, A. D., and Sun, Q.: Determining the Anthropogenic Greenhouse Gas 575 Contribution to the Observed Intensification of Extreme Precipitation, Geophysical Research Letters, 47, e2019GL086 875, https://doi.org/10.1029/2019GL086875, 2020.

Sun, Q., Zwiers, F., Zhang, X., and Yan, J.: Quantifying the Human Influence on the Intensity of Extreme 1- and 5-Day Precipitation Amounts at Global, Continental, and Regional Scales, Journal of Climate, 35, 195 – 210, https://doi.org/10.1175/JCLI-D-21-0028.1, 2022.

---

## Author Response (AR1)

Dear Authors,

Following your thorough replies to the comments provided by both Reviewers, I would like to invite you to prepare a revised version of your study. Amongst other edits, I would encourage you to provide a thorough contextualisation of your work along the lines of your replies to Reviewer #1, and a more intuitive illustration of your methodology along the lines of your replies to Reviewer #2. As ESD is a journal with a broad readership from different fields of Earth Science, it is important that your study does not only speak to specialists in a specific sub-field. Make a careful evaluation of which edits should be included in the main text and which in the Appendix or SI, as the main paper should ideally be accessible to the ESD readership without needing extensive reference to appendices and supplements.

Best Regards,
Gabriele Messori

Dear editor,

Thanks a lot for your managing of our manuscript. We have implemented changes to address the reviewers' comments. Below, we provide a response to both reviewers with short comments on the way in which we addressed the comments. For the longer replies, please refer to the initial responses.
We are aware that there still are a significant number of references to the supplementary information, and the amount of material in supplementary information is considerable. Nonetheless, we are confident that this all is indeed strictly supplementary and that the manuscript in itself is stand-alone, since none of the main conclusions depend on any of the additionally provided analyses or considerations.

Best,
Iris de Vries, on behalf of all authors

Comments of Referee #1

This paper proposes a ridge regression approach to the detection and attribution of externally forced changes in mean and extreme precipitation. This is an interesting idea that certainly merits exploration, but before devoting a lot of time to understanding the details of the paper and the results that are obtained, I think it is necessary for the authors to better explain their method and to situate it within the pantheon of methods that are already available for detection and attribution.

Ridge regression is a technique that "regularizes" regression problems, such as that described in equation (1) of the paper, in which the predictor variables contained in matrix X are multicollinear. In the generalized least squares formulation of the regression used in detection and attribution this matrix is composed of model simulated estimates of the responses to external forcing in the form of space-time patterns of change. Depending on variable, period considered, domain of interest and how data are processed, the expected space-time patterns of responses to different forcing factors (often called fingerprints) can be strongly correlated, which results in a regression "design matrix" X that may be ill conditioned. Ridge regression is a technique that can be used to overcome this problem, although I imagine at the cost of introducing some bias into the estimated signal scaling coefficients β. Note that referring to these coefficients as "fingerprints" seems unusual to me.

The concept of regularization, however, also arises in a second way in the detection and attribution problem. Considering again equation (1), the generalized least squares approach (and also its total least squares extension) requires knowledge of the variance covariance matrix of the residuals ε, which are regarded as resulting from natural internal climate variability. Thus, the variance-covariance matrix is generally estimated from unforced control simulations, using as many climate-model simulated realisations of ε as possible. Even though many climate-model simulated realizations of ε are now generally available, the estimated variance-covariance matrix may not be of full rank or may remain uncertain. Thus, it is also often regularized, using an approach similar to the regularization used in ridge regression, but applied to the noise term rather than the signal term of equation (1). See Ribes et al (2013a, doi:10.1007/s00382-013-1735-7, and 2013b, doi:10.1007/s00382-013-1736-6). Presumably one would want to regularize both aspects of the problem, and also take signal uncertainty into account as is done in the total least squares approach to the regression problem (see again Ribes et al., 2013a and 2013b, and also Allen and Stott, 2003, doi:10.1007/s00382-003-0313-9).

How the combined model represented by equations (1-3) relates to existing techniques, and now the noise that results from internal variability comes into play and is accounted for in their subsequent application in the paper is not made clear, and I think should be clarified before results can be considered.

We would like to thank the reviewer for the important remarks and suggestions. Below we outline in brief comments how we addressed the comments. For the extensive reply to the comments we refer to the initial response to reviewer 1.

A condensed version of the reply in the initial response to reviewer 1 has been added to the methods section – ranging from **L86 to L115** – to explain our method in more detail, and explicitly relate our method to existing and upcoming methods, and hopefully make the differences more clear. In addition, a flow chart has been added – see also our response to referee #2's comments – which should further clarify the ridge regression part of the method.

Also, I think it is necessary for the authors to discuss whether the proposed methods, which basically use linear statistical models that therefore implicitly assume Gaussian, or near Gaussian errors, are suitable for the data to which they are applied. Indicators of extreme precipitation, such as Rx1day at individual grid boxes, are certainly not Gaussian.

There is no need for the predictors of Rx1d trends to be normally distributed in our method, as explained in the first response to reviewer #1, and refer the reviewer to the online reply for a more extensive justification.

A final general comment is that the relatively heavy of use of acronyms in this paper is not very reader friendly.

We have removed the acronyms for forced response (FR) and forced response estimate (FRE), which leaves Rx1d, PRCPTOT, ridge regression (RR), signal-to-noise ratio (SNR) and empirical orthogonal function (EOF). Given that these are commonly used acronyms and/or concern the essence of our study (RR), we think this is a manageable collection of acronyms.

Comments of Referee #2
Short responses, please refer to the initial response to reviewer 2 for extensive replies.

Overall comments:

This study conducts a signal detection analysis for global changes in mean and extreme precipitation using three observational datasets and CMIP6 multi-model outputs. The authors apply a ridge regression (RR) method to construct fingerprints, which helps increase a signal-to-noise ratio of precipitation change patterns. Results show a robust detection of anthropogenic signals in all observations for both mean and extreme precipitation even when removing global mean trends, further supporting the human-induced intensification of global hydrological cycle. I find this paper very well written with sufficient details provided about methods as well as various sensitivity tests and therefore suggest publication after addressing some minor issues.

Thanks again for your comments and efforts. We list the changes made below, and refer to the initial response to reviewer 2 for extensive replies.

Major comments:

1. Although method details are provided, it would be useful to explain more clearly what are benefits of the attribution approaches employed, including ridge regression, EOF-based metric for target variable, and GMST-based signal estimation. All of these procedures seem to contribute to increase signal-to-noise ratio but how they do and what step is more important. The authors provide some associated results from sensitivity tests but an overall explanation of their method possibly with a schematic would be helpful for readers to understand the contribution of each step to the final signal detection.

We added additional explanation of the method (**L86-L115**) to the methods section, as well as a flowchart **(Figure 1)** explaining the ridge regression steps. A flowchart addressing the EOF-based targets is added to the supplementary information (SI Figure S1).

In addition, we added a section to the supplementary information, **SI Section S2.4**, to provide additional information on the effects of design choices on the signal-to-noise ratio/time of emergence. The crucial points of this SI section, namely the effect of regularisation, were already discussed in the original manuscript, section 3.4.

2. An important motivation of considering different periods and datasets is opposing conclusions by previous studies about model overestimation or underestimation of the observed trends. I am wondering if the authors can go further and compare their results with some previous studies. For instance, if studies based on the latter half of 20th century trends find model underestimation, the authors can assess their model trends for the same/similar periods. Another point here is that the present study uses absolute units of precipitation while most of previous studies considered relative changes or aggregated values. It would be good to discuss possible influences of this difference.

We addressed the comparison of different precipitation metrics in supplementary **section S2.3**, and refer to here in the manuscript **L377** and **L483**.

We addressed the comparison to previous studies by adding the table below to the supplementary info. This systematic comparison to other studies has also led to the addition of a few references throughout the manuscript.

Previous studies report results ranging from model under- to model overestimation of observed trends for all trend periods and units (absolute vs. normalised). There is no systematic explanation for why opposing results are found. Therefore, we do not go further than stating this, in **L482-492**.

**PRCPTOT**

| Paper | Model archive | Obs dataset | Spatial region | Variable | Method | Trend periods | Models w.r.t. observations? | Remarks |
|---|---|---|---|---|---|---|---|---|
| Noake et al. (2012) | CMIP3 | GHCN, CRU, VASClimO | Global land, separated into 5deg latitude bands. Scaling factors determined for spatiotemporal aggregate, not per latitude band | Seasonal PRCPTOT percentage change per latitude band | Optimal fingerprinting | 1952-1990, 1960-1999, 1951-1990, 1975-1999 periods | - | "-" applies to scaling factor best estimate for seasons and observational datasets in which significant change is detected (confidence interval does not include 0), and holds for all trend periods |
| Wu et al. (2013) | CMIP5 | GHCN | Northern hemisphere land | PRCPTOT percentage change | Optimal fingerprinting | 1952-2011 | - | |
| Polson et al. (2013) | CMIP5 | GHCN, CRU, VASClimO, GPCC | Global land, separated into 5deg latitude bands. Scaling factors determined for per latitude band aggregate, not per latitude band | Seasonal PRCPTOT percentage change per latitude band | Optimal fingerprinting | 1951-2005 (2000 for VASClimO) | - | Applied Noake's method to CMIP5, "-" applies to scaling factor best estimate for seasons and observational datasets in which significant change is detected (confidence interval does not include 0), GPCC never shows a detectable climate signal. |
| Fischer & Knutti (2014) | CMIP5 + CESM initial condition ensemble | HadEX2, GHCNDEX | Global | Spatial distribution of gridpoint trends in PRCPTOT, expressed in terms of local sigma (based on 1986-2005 interannual variability) | Spatial probability distribution comparison | 1960-2010 | + | Models estimate more regions with positive trends in PRCPTOT, but not enough negative trends --> too much wetting |
| Knutson & Zeng (2018) | CMIP5 | GPCC | Global, per gridpoint | Linear trend in PRCPTOT | Linear trend fitting to grid point timeseries | 1901-2010, 1951-2010, 1981-2010 | - | Models cannot produce the magnitude of positive nor negative trends in obs. Discrepancy gets stronger in later trend periods |

**Rx1d**

| Paper | Model archive | Obs dataset | Spatial region | Variable | Method | Trend periods | Models w.r.t. observations? | Remarks |
|---|---|---|---|---|---|---|---|---|
| Min et al. (2011) | CMIP3 | HadEX | NH land, separated into (overlapping) regions: mid-latitudes, tropics | Rx1d and Rx5d Probability index: 0-1 quantile per value, based on fit GEV per gridpoint | Optimal fingerprinting | 1951-1999 | - | "-" applies to scaling factor best estimate for regions where there is detection |
| Zhang et al. (2013) | CMIP5 | HadEX2 + russian station data | NH land, separated into (overlapping) regions: Western Eurasia, Eastern Eurasia, North America, mid-latitudes, tropics | Rx1d and Rx5d Probability index: 0-1 quantile per value, based on fit GEV per gridpoint/station and then interpolated | Optimal fingerprinting | 1951-2005 | 0/+ | Scaling factor estimates include 1, but best estimates are still below 1 |
| Fischer & Knutti (2014) | CMIP5 + CESM initial condition ensemble | HadEX2, GHCNDEX | Global | Spatial distribution of gridpoint trends in Rx5d, expressed in terms of local sigma (based on 1986-2005 interannual variability) | Spatial probability distribution comparison | 1960-2010 | - | Models don't show a large enough land fraction exhibition positive trends, and do not reproduce the magnitude of the largest trends seen in observations |
| Fischer & Knutti (2016) | CMIP5 and EURO-CORDEX | E-OBS/Ensembles | Europe | Changing occurrence of historical >90 percentile values of daily precipitation | Probability distribution comparison | 1951-1980 and 1981-2013 distributions | - | Models show smaller increase in intensity of >90th percentile daily precipitation amounts |
| Borodina et al. (2017) | CMIP5 + CESM initial condition ensemble | GHCNDEX, HadEX2 | Global land, selected wet regions only (wettest 40%, agreed across models) | Rx1d percentage change per gridpoint as a function of GMST [%/K], averages over wet regions, as well as land area fraction experiencing positive Rx1d trends | Trend comparison | 1951-2005 | - | Models show smaller trends than both observational datasets, but HadEX2 shows smaller trends than GHCNDEX |
| Paik et al. (2020) | CMIP5 | HadEX2 | Global land, separated into (overlapping) regions: Western Eurasia, Eastern Eurasia, North America, mid-latitudes, tropics, wet and dry regions. | Rx1d and Rx5d Probability index: 0-1 quantile per value, based on fit GEV per gridcell/station and then interpolated. Scaling factors | Optimal fingerprinting | 1950-2020 | 0/+ | "0" applies to EU and dry regions, where models and observations agree. For all other regions with detection, models overestimate the change ("+") |
| Paik et al. (2020) | CMIP5 | HadEX2 | Global land, separated into (overlapping) regions: Western Eurasia, Eastern Eurasia, North America, mid-latitudes, tropics, wet and dry regions. | Rx1d and Rx5d Probability index: 0-1 quantile per value, based on fit GEV per gridcell/station and then interpolated. Spatially averaged trends, normalised by GMST | Trends in %/K | 1950-2020 | - | Note: same study as above. In all regions where forced change is detected, models underestimate observations when trends are assessed. In these same regions, scaling factors suggest that models overestimate change. |
| Sun et al. (2022) | CMIP6 and CanESM2 LE | HadEX2 stations + russian and chinese station data | Global, continental, regional | Rx1d and Rx5d, Non stationary spatiotemporal varying GEV-based optimal fingerprinting, no normalisation: absolute units of precipitation (log) | Non-optimal variant of optimal fingerprinting: scaling factor determination but no internal variability covariance corrections | 1950-2014 | + | "+" applies to all continents/regions, and also global level, but Northwestern Europe (Scandinavia/UK) where scaling factors are around 1 ("0") |

Table 1: Previous D&A studies on PRCPTOT and Rx1d, including their main findings on whether modelled forced changes are smaller (-), similar (0) or larger (+) than observed forced changes

3. The lower detectability in GHCNDEX observations are suggested to be due to the poorer spatial coverage. Regarding this issue, I would suggest using Rx5d. As I understand, Rx5d has larger spatial coverage than Rx1d and comparison with Rx1d-based results may provide a way to support the authors' interpretation. Another way would be to compare detection results from using a selected model run but with different spatial coverages applied.

We looked into this, but found that Rx5d does not provide higher coverage. Because we want to keep the difference between mean and extreme precipitation measures as large as possible, (see initial response to reviewer 2) we maintain Rx1d as the metric for extreme precipitation.

Minor comments:

L8: Indicating analysis period or trend period with signal detection would be useful here.

Changed to "[...] to assess the degree of forced change detectable in the real-world climate *in the period 1951-2020*." in **L8.**

L17-19, L58-64: Better comparisons can be made by applying the same periods as those used in previous studies. See my major comment above.

See reply to major comment above: both previous studies as well as we assess multiple trend periods. Disagreements across studies and observational datasets remain.

L20-21: Is this confirmed by repeating detection analysis using NH-extratropics only?

Yes, see supplementary information, SI section S3.

L34: "discrepancies with respect to observations". Its meaning is unclear.

Changed the sentence to "There can also be discrepancies between model representations of the water cycle and observations." in **L33-34**.

L69-71: Need to explain what the previous studies have found additionally using these "data-science methods". Also, what's the novelty of this study compared with them? Is it detection based on spatial pattern information alone?

This has now been added to the method section, from **L87** we compare to other D&A methods in general, and from **L111** we specifically refer to other data-science methods.

L108-109: "Trend biases due to this structural difference ... negligible". But the cited reference considered south-east Australia only?

In **L138** we have added the reference from the previous sentence, which makes this statement in a general context.

L201: How to define S when global means are removed?

We do not think additional explanations in the paper are required here, since the forced trend is still the predicted variable in this case, which is the basis for S.

L212: "CMIP6 ssp245" should be "CMIP6 historical"?

This has been changed, **L249.**

L227: "virtually identical". adding spatial correlation would help with this.

This has been added to **L264-265.**

L314-316: This suggests possible dependence of Rx1d FRE on temperature, resembling global warming slowdown due to PDO influence?

Potentially, although we do not have enough evidence to claim that the levelling off of the trends is not simply due to shorter trend length and internal variability. Attributing changes in trend slope to lange scale modes of variability is outside the scope of this study. Hence we have not added statements on this to the manuscript.

L331-332: "results … hold when the global mean is used as FR target". Then what are benefits of using EOF-based metric for target variable?

See the discussion in the initial response to reviewer 2. We have added to the method section to further justify this choice of target in **L169-172**.

L382-383: "accuracy of the CMIP6 climate models in simulating the processes …". It's unclear how the authors get this conclusion. Observation-model agreement in residual variability? More explanation would be useful.

From **L428** we changed this to "Taken together, the above shows, first, detection of forced change in mean and extreme precipitation beyond a global mean trend, and second, the power of RR for signal extraction from high-dimensional noisy data. Finally, the fact that the relationship between relative spatial precipitation patterns and the forced precipitation trend derived from climate model simulations (the ridge model) holds in observations, suggests accuracy of the CMIP6 climate models in simulating the processes relevant to the spatial pattern of forced change in mean and extreme precipitation."

L394-395: "(not shown)". This looks important and I suggest showing them in the supplement.

Added in SI section **S2.4**

L428: "value of RR-based fingerprint construction". What happens in detection or SNR without applying RR? See my major comment above.

See the additions to the supplementary information and responses to other comments. In the manuscript, **L209-213**, unchanged text, the effect of regularisation was already described, as well as in section 3.4, from **L457** onwards.

**References in table**

Noake, K., Polson, D., Hegerl, G., and Zhang, X.: Changes in seasonal land precipitation during the latter twentieth-century, Geophysical Research Letters, 39, https://doi.org/10.1029/2011GL050405, 2012.

Wu, P., Christidis, N., and Stott, P.: Anthropogenic impact on Earth's hydrological cycle, Nature Climate Change, 3, 807–810, https://doi.org/10.1038/nclimate1932, 2013.

Polson, D., Hegerl, G. C., Zhang, X., and Osborn, T. J.: Causes of Robust Seasonal Land Precipitation Changes, Journal of Climate, 26, 6679 – 6697, https://doi.org/10.1175/JCLI-D-12-00474.1, 2013.

Knutson, T. R. and Zeng, F.: Model Assessment of Observed Precipitation Trends over Land Regions: Detectable Human Influences and Possible Low Bias in Model Trends, Journal of Climate, 31, 4617 – 4637, https://doi.org/10.1175/JCLI-D-17-0672.1, 2018

Min, S.-K., Zhang, X., Zwiers, F. W., and Hegerl, G. C.: Human contribution to more-intense precipitation extremes, Nature, 470, 378–381, https://doi.org/10.1038/nature09763, 2011.

Zhang, X., Wan, H., Zwiers, F. W., Hegerl, G. C., and Min, S.-K.: Attributing intensification of precipitation extremes to human influence, Geophysical Research Letters, 40, 5252–5257, https://doi.org/10.1002/grl.51010, 2013

Borodina, A., Fischer, E. M., and Knutti, R.: Models are likely to underestimate increase in heavy rainfall in the extratropical regions with high rainfall intensity, Geophysical Research Letters, 44, 7401–7409, https://doi.org/10.1002/2017GL074530, 2017.

Paik, S., Min, S.-K., Zhang, X., Donat, M. G., King, A. D., and Sun, Q.: Determining the Anthropogenic Greenhouse Gas 575 Contribution to the Observed Intensification of Extreme Precipitation, Geophysical Research Letters, 47, e2019GL086 875, https://doi.org/10.1029/2019GL086875, 2020.

Sun, Q., Zwiers, F., Zhang, X., and Yan, J.: Quantifying the Human Influence on the Intensity of Extreme 1- and 5-Day Precipitation Amounts at Global, Continental, and Regional Scales, Journal of Climate, 35, 195 – 210, https://doi.org/10.1175/JCLI-D-21-0028.1, 2022.

---

## Author Response (AR2)

Reviewer comments
Replies

First, I would like to thank the authors for their detailed responses to my comments and for the improvements that have been made to the paper, which in my view, is much improved, particularly regarding the presentation of key concepts that are needed to understand the paper and how it is situated in the "D&A" literature.

I do have some additional specific comments that might help to further improve an already very good paper, which I hope you find useful.

Thank you very much for this positive evaluation of our revised manuscript, and we greatly appreciate the effort and conscientiousness you put into reviewing this version so thoroughly once again. Below, we provide point-by-point answers to your additional comments.

60-77: How does this paper help to resolve the issues that are raised here concerning the robustness of findings from D&A studies and the representation of precipitation related processes in models? The text that begins with "Here we show …" (line 70) doesn't really answer that question.

We updated the text in L70-76 to answer this question more explicitly.

116-119: It seems that you've tried to create a very "balanced" climate model dataset (3 runs per model, 450 years of PI-control per model for as many models as possible). A few words on the sampling approach (e.g., what motivates it, what problems are avoided through a balanced approach, and what other problems might arise – see comments below) might be appropriate.

That is true, we added some words on the motivation for the model data selection in L119-126.

134-135: I think the crucial issue is not so much the locations that grid-cells represent, but rather, the quantities that they represent. The question of what quantity they represent is easier to understand in the case of Rx1day. I think we can all agree that the Rx1day values obtained from models can only be interpreted as annual maxima of daily spatial mean (grid box average) precipitation amounts. In contrast, when a product like HadEX3 is produced from station data, Rx1day is first determined at stations and then those point Rx1day values are spatially interpolated to obtain a grid box value. This is a different number from the annual maximum of daily grid box averages of 1-day point precipitation amounts, which is how one would interpret model output. That is, there is an order-of-operations difference that could potentially influence model-observations comparisons. If we had dense observational coverage (e.g., say at least 50 rain gauges in each model grid square), we could presumably first calculate the grid-box average precipitation amount, and then calculate Rx1day from those averages, bringing us conceptually closer to the thing that the models produce.

Indeed, we agree that we did not state this clearly enough, and made this more explicit now. Just to add: it is an inconvenient fact that models output grid cell mean quantities, whereas

observational Rx1d values are not determined from gridded station data (and even if they were, it would still not be the same since gridded observational PRCPTOT data does not reflect the grid cell mean), but from raw station data. We did our best to not make this discrepancy worse by ordering our preprocessing steps on model data in the way that best reflects the observational order of operations. Namely, we first 'extract' the Rx1d values at each individual gridpoint (~equivalent to determining station maxima), and then regrid onto the coarser observational grids (~equivalent to regridding). Fact of the matter is, however, that the maximum of a "small" grid cell on the native model grid still differs greatly from individual station maxima.

We've rewritten this paragraph, L141-151, and also added a sentence or two to section 2.2 to underline the importance of order of operations again, L154-156.

163-165: The paper seems to be getting ahead of itself a bit here. The results show this to be true for the precipitation variables you consider, but at this stage, it isn't known whether we should expect the same for precipitation as for surface air temperature.

That's a valid remark. Given that the linear trendmaps in Fig. 2a/b have high spatial correlation with the first EOF pattern (visible in Fig. 2 and Supp Fig. S2), and global mean timeseries with the first PC (visible in Supp Fig. S3) we can deduce that external forcing is represented in the first EOF. We shuffled the text around a bit to make this point in the correct logical order, see L179-189.

174: It seems clear from Fig. S2 that you have two groups of models with a substantial gap in the range of sensitivities that isn't sampled. It would be worth including some discussion of how that heterogeneity could potentially have affected results.

This gap is remarkable indeed (for Rx1d) and is related to the different warming rates of the models (clausius-clapeyron). Although there is an unsampled space between the two clusters, the ridge model would still end up somewhere in the middle (which lies in the non-sampled space) to minimise the bias in both directions. Supp. Fig. S4 shows and main text L246 states that the RR models have low sensitivity to any left-out model (pre- versus post-crossvalidated performance is nearly identical). This indicates that the predictors of all-but-1 models can train an RR model that generalises well to the left-out model.. Furthermore, we show in supp sect. S2.3 and state in the main text that the results are not fundamentally affected by differing climate sensitivities, since the results do not change if we normalise each model's forced response by its temperature change.

This leads us to think that the heterogeneity does not affect the results much, given that there is sufficient data on both sides of the gap. We think our referral to the pre- and post crossvalidation results, and the results where the dependence on climate sensitivity is removed by normalising wrt temperature change, provide enough proof of robustness, and we do not add additional discussion on this.

200: Is the cost function minimized separately at each location? In minimizing the cost function, how do you account for the impacts of spatial and temporal dependence?

The cost function is minimised globally across the masked coverage), see eq. 4 in the paper.

There is no location-dimension in the two variables (y and Xbeta) in the cost function which govern the magnitude of the error. However, we fully agree with the reviewer that there is (strong) spatio-temporal depedencies between the predictor variables in the regression formulation. Because of that, and the large amount of predictors and data points, there would be high (perhaps inevitable) risk of overfitting in a standard OLS multiple linear regression framework. Preventing this – overfitting due to spatial dependence and large data matrices – is exactly the goal of regularisation key to ridge regression (see also more detailed description in L232-236). Because of the spatial dependence in the data, predictors are not orthogonal and OLS regression would lead to non physical and high coefficients to, in a way, "force" an approximation of the target. This is overcome by the L2-norm regularisation used in ridge regression, since this forces coefficients to become more homogeneous: due to the squared-sum penalty, high coefficients are penalised most strongly, effectively resulting in moderate coefficients, which, when applied to geophysical data with spatial dependence, leads to spatially coherent maps of coefficients. Hence, regularisation accounts for spatial correlation in the data.

Temporal dependence in the data is induced through long-term trends in the forcing, or decadal variability in the climate system. In the way we applied ridge regression, autocorrelation in the data does not affect overfitting in the same way as spatial correlation does, since every year makes up a separate equation in the linear system. We use a careful cross-validation scheme, in which we strictly separate the fitting of the model (on a set of $k$ climate models) from the application to observations, or other, unseen climate models. This has the effect that the algorithm has no knowledge of the temporal variability in the application model or in observation. This ensures that only the temporal structures where the models agree (i.e. those that are not due to internal variability or model biases) are reflected in the coefficient fingerprint.

We mention the role of regularisation in suppression spatial artefactss in L232-236, and the temporal aspects are addressed in L223-228.

269-271: Since the fingerprints are model based, they could presumably be shown for the global domain, with outlines of the regions with observational coverage. But perhaps this suggestion is naïve (see my previous comment).

Even though the fingerprints are model based indeed, and the model has full coverage, we still do not (cannot) determine the fingerprints for the full coverage since our ultimate goal is to apply the fingerprints to observations. We want to predict/estimate the forced response from observations, which means that we need an RR model that maps the *observed* gridcells to a forced response estimate. The RR-model is thus trained only with observed grid cells.

If we would determine the weights based on global coverage (i.e. a coefficient for a predictor at every grid cell), and only use a part of the coefficients when we apply the RR model to observations (thus leaving out a large fraction of the predictors), the result would not predict the forced response anymore. Hence, the masked fingerprints we show, show the full RR model.

If the cost function is minimized "globally" across the observation grid, then a question that emerges would be whether the geometry of the observation grid affects the fingerprints that are obtained, and what implications that might have for detection (or the timing of detection).

As the cost function is minimised across the observational grid, the weights depend on the coverage and details of the grid: unobserved regions can not contribute to the forced response estimate. We refer to the consequences of this several times in the paper, pointing out the lower coverage of GHCNDEX, and the dominance of the Northern hemisphere due to observational coverage (and land mass) being much larger there, e.g. in L517-521.

308-309: I think this is a consequence of having a collection of models that do not sample the model sensitivity spectrum very well. (See also my comment concerning line 174).

The differences in climate sensitivity, and the unsampled climate sensitivities you pointed out above, may contribute to the fact that the ridge model does not perfectly predict the modelled forced response from model data. However, the main reason for the phenomenon these lines refer to – the flattening of forced response estimates relative to the targets – is the use of regularisation. We show this with the plots below:

[Figure]

Figure 1: Forced response estimates obtained by applying the RR model **with regularisation parameter lambda_min** to model data (shading) and observations (colours), PRCPTOT (left) and Rx1d (right) – same as Fig. 3c and d in the main paper, apart from regularisation parameter choice.

These plots show the forced response estimates made by a minimally regularised RR model (lambda_min, as described in the supplementary information). Effectively, a very small regularisation parameter means that the overfitting is not much reduced, and therefore the variance in the estimates is still large, and there is little bias. We see that the shading in these plots is centred on the black line, and does not show the "flattening" effect we described in L308-309 (of the previous manuscript, referred to by the reviewer).
The more we regularise, the more we dampen variance, but this comes at the cost of flattening the trend (which is also a source of variance): in the limit, infinite regularisation leads to a ridge model with spatial coefficients of zero, and just an intercept, predicting the mean of the timeseries, as shown below:

[Figure]

Figure 2: Forced response estimates obtained by applying the RR model **with regularisation parameter very large** to model data (shading) and observations (colours), PRCPTOT (left) and Rx1d (right) – same as Fig. 3c and d in the main paper, apart from regularisation parameter choice.

We see a mild signature of this bias-variance trade-off regularisation effect if we choose the lambda that is optimal for generalisability and interpretability (lambda_sel), which is what we described in L308-309.

360: Confidence in the consistency of results perhaps? I'm not sure what it means for a method to be consistent.

This was sloppily formulated indeed. What we mean is that the method produces consistent results in different contexts and is therefore robust. So we have rephrased this to "This consistency of results increases confidence in the robustness of the method, …", L382-383.

387: I'm not sure I understand what it means for change to be in phase, but of opposite sign.

It means that the correlation is -1 → the temporal evolution is the same, but where the forced response grows in the positive direction over time, the changes in these regions become increasingly strongly negative. We added a clarification in L410.

393: Replace "… coefficients flip sign." with "… coefficients that flip sign."
Done.

495-496: There are certainly papers that use precipitation datasets with greater coverage, but it takes a lot of work and personal contacts to collect that data and organize it for use in a D&A study. But even with that kind of work, coverage over land is still not very good. I wonder if we will get to the stage where we might have enough trust in reanalyzed precipitation to use it in a D&A study?

As you say, station data has certain "irreducible" caveats, such as being point-measurements, sparsely distributed, and sometimes even subject to (geo)political considerations. Also reanalysis data has its caveats, as you hint at. Nonetheless, reanalysis precipitation data might be fit for lower-resolution, longer timescale purposes, for example,

Bonfils et al. (2020) successfully use reanalysis data in a pattern correlation D&A study (assessing temperature, precipitation, and aridity jointly, i.e. not solely precipitation based). In the same way, station data also can be fit for different purposes, despite its imperfections. There is value in different lines of evidence and different types of detection and attribution studies and statements. Besides, we can learn many things from detection efforts that are not set out to detect the strongest signal, but that aim to combine physical understanding of the manifestation of the signal with the signal strength. The question whether forced climate change is detectable in sparse station observations is not solely aimed at making the best detection statement possible, possibly reanalysis would yield much more constrained results. The question is whether this type of very direct, "true" observations can be used in a physically explainable manner to show the effects of external forcing.

Anyway, to prevent getting lost in (interesting!) philosophical contemplations, a short answer to the question: we can probably learn things about climate change using reanalysed precipitation if we ask the right questions for the possibilities reanalysis data provides.

**References**

Bonfils et al. (2020). https://doi.org/10.1038/s41558-020-0821-1